# ATTS: Asynchronous Test-Time Scaling via Conformal Prediction

**Jing Xiong**[1][*], **Qiujiang Chen**[1][*], **Fanghua Ye**[4], **Zhongwei Wan**[2], **Chuanyang Zheng**[5]
**Hui Shen**[1], **Chenyang Zhao**[2], **Hanbo Li**[3], **Chaofan Tao**[3], **Haochen Tan**[3]
**Haoli Bai**[3], **Lifeng Shang**[3], **Lingpeng Kong**[1], **Ngai Wong**[1]

[1]The University of Hong Kong    [2]Independent Researcher    [3]Huawei Technologies Co., Ltd
[4]University College London    [5]The Chinese University of Hong Kong

 https://github.com/menik1126/Asynchronous-Test-Time-Scaling

## Abstract

Large language models (LLMs) benefit from test-time scaling but are often hampered by high inference latency. Speculative decoding is a natural way to accelerate the scaling process; however, scaling along both the parallel and sequential dimensions poses significant challenges, including substantial memory-bound execution and synchronization overhead. We introduce ATTS (Asynchronous Test-Time Scaling), a statistically guaranteed adaptive scaling framework that follows the hypothesis testing process to address these challenges. By revisiting arithmetic intensity, ATTS identifies synchronization as the primary bottleneck. It enables asynchronous inference through online calibration and proposes an ordinal classification algorithm that supports a three-stage rejection sampling pipeline, scaling along both the sequential and parallel axes. Across experiments on the MATH, AMC23, AIME24, and AIME25 datasets and across multiple draft–target model families, we show that ATTS delivers up to *56.7x* speedup in test-time scaling and a *4.14x* throughput improvement, while maintaining accurate control of the rejection rate, reducing latency and memory overhead, and incurring no accuracy loss. By scaling both in parallel and sequential dimensions, we enable the 1.5B/70B draft/target model combination to achieve the performance of the state-of-the-art reasoning model o3-mini (high) on the AIME dataset.

## 1 Introduction

With the rapid advances in large language models (LLMs), attention is increasingly turning to *reasoning models* (Guo et al., 2025; Muennighoff et al., 2025; McCoy et al., 2024; Shao et al., 2024)—systems that transcend next-token prediction in order to emulate human-like reasoning behaviors. These models excel at leveraging complex reasoning chains, especially in test-time scaling settings (Snell et al., 2024; Li et al., 2025; Muennighoff et al., 2025; Zeng et al., 2025), and have shown strong potential in mathematical reasoning (Xiong et al., 2022; 2023b;a).

Test-time scaling (Chen et al., 2025; Muennighoff et al., 2025; Guo et al., 2025) constitutes a new paradigm that enhances the model's reasoning capabilities by allocating additional computational resources during the inference stage. Typically, test-time scaling can be categorized into two approaches: sequential scaling (Muennighoff et al., 2025; Guo et al., 2025) and parallel scaling (Chen et al., 2025). However, despite its potential, the challenge of efficiently managing increasing sampling size or complexity during inference remains a critical limitation, hindering the achievement of high-performance deployment.

Benefiting from the shared-prefix mechanism of the inference engines (Kwon et al., 2023; Zheng et al., 2024), parallel scaling (Chen et al., 2025) increases the number of samples concurrently, thereby partially mitigating the inference-time latency and memory footprint introduced by scaling the per-trajectory token budget (i.e., longer reasoning paths), while simultaneously improving token-sampling throughput.

Although some methods (Huang et al., 2025; Wan et al., 2024) that adopt confidence-based early stopping of reasoning chains improve parallel sampling efficiency, problems still remain in memory

---

[*]Contact Email: junexiong@connect.hku.hk, Equal contribution.

efficiency and high inference latency. Another potential issue is that early stopping prunes away potentially correct reasoning paths and reduces the diversity of the output space.

Speculative decoding (Li et al., 2024a; Leviathan et al., 2023; Kim et al., 2023; Pan et al., 2025b; Yang et al., 2025) represents a promising approach for accelerating decoding. In this framework, a lightweight draft model generates tokens, which are subsequently validated and refined by a target model. This dual-phase approach not only speeds up inference by offloading most of the generation process to the draft model, but also ensures that the final outputs retain high fidelity, thereby achieving a favorable balance between efficiency and accuracy.

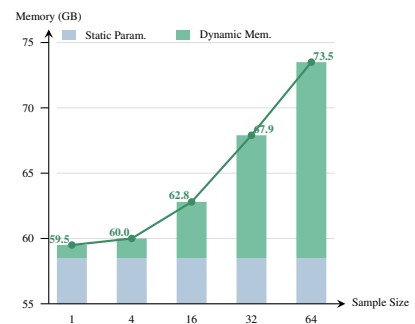

Figure 1: Memory Overhead vs. Sampling Sizes (QwQ 32B, Token Budget 500)

However, when speculative decoding (Pan et al., 2025b; Yang et al., 2025) meets test-time scaling, the decoding process faces *two key challenges*. The first is the *memory bottleneck* of the target model during the prefill phase. As shown in Figure 1, as the number of sampling increases, the memory overhead of the target model tends to grow due to KV cache accumulation. This effect becomes more pronounced in target models when attempting to scale the number of requests from the draft model. During real-world deployment on the SGLang server (Zheng et al., 2024), high-concurrency sampling, especially when simultaneously validating multiple long reasoning chains, can lead to memory peaks that easily exceed the GPU's maximum capacity, causing the server to crash. Therefore, it is crucial to constrain the request budget from the draft model to the target model within a manageable range.

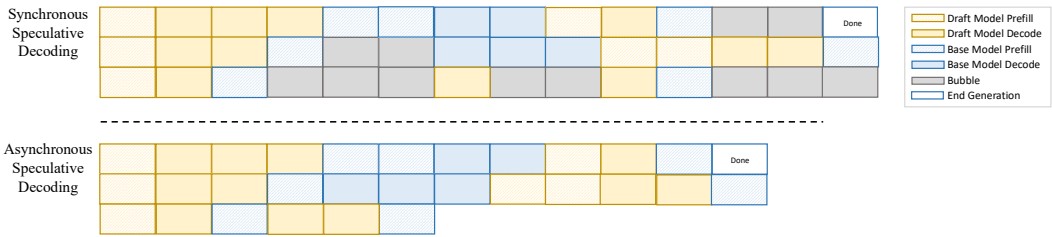

Figure 2: Comparison of naive and asynchronous speculative decoding.

In speculative decoding, one stage involves rejection sampling: prior to acceptance, the target model either ranks draft-generated candidates or computes a divergence between the draft and target distributions, introducing an additional *synchronization overhead bottleneck*. As illustrated in Fig. 2, during the multi-turn sampling process (with a sampling quantity of 3 at each turn), if the target model aims to reject the sampling with the lowest two confidences, a ranking operation must be performed at each turn. Given the limited computational and memory resources, the target model needs to prioritize processing the most important requests. Especially in test-time scaling, when combining sequential and parallel scaling, the synchronization overhead from precise budget control and the pursuit of globally optimal ranking is amplified. Although this issue has been widely discussed in the context of tool calls (Gim et al., 2024; Ginart et al., 2024), it has not been formally proposed in the context of test-time scaling.

To analyze the synchronization bottleneck and address the two challenges mentioned above, we first introduce a novel variant of arithmetic intensity called *asynchronous arithmetic intensity* to analyze the system bottleneck, and then explore conformal prediction (Vovk et al., 2005; 2003; Romano et al., 2020; Lei et al., 2018) for ranking predictions to design the asynchronous algorithm. In our formulation, conformal prediction defines a prediction set $C_\alpha$; sampling in $C_\alpha$ are rejected, while sampling outside are accepted. This yields a distribution-free guarantee that the right sampling is retained (i.e., lies outside $C_\alpha$) with high probability, enabling asynchronous test-time scaling. This paper presents the following contributions:

- We propose *asynchronous arithmetic intensity*, a performance metric designed to character-ize and quantify throughput/latency bottlenecks that emerge in test-time scaling scenarios.

- We introduce *conformal prediction* to tackle prediction ranking, and—leveraging the resulting ranking—construct stable prediction sets that mitigate GPU-memory bottleneck risks.

- We propose ATTS, a training-free, lossless acceleration method that achieves a *56.7x* speedup in test-time scaling and a *4.14x* throughput improvement in both sequential and parallel settings.

## 2 PRELIMINARY

We first introduce how to build the prediction set in the classical setup, and then present our setup.

**Classic Setup.** Formally, let

$$D_{\text{cal}} = \big( (X_1, Y_1), \ldots, (X_n, Y_n) \big) \tag{1}$$

denote the calibration dataset. Each pair $(X_i, Y_i)$ for $i = 1, \ldots, n$ is a data point, consisting of an $X_i$ (the input question for the $i$-th example) and a ground truth denoted as $Y_i$. The symbol $n$ denotes the size of the calibration dataset. For each input $X_i$, we draw $m$ candidate sampling

$$(\hat{Y}_i^1, \ldots, \hat{Y}_i^m), \tag{2}$$

where $m$ is the number of samples per input (the sampling budget), and $\hat{Y}_i^k$ denotes the $k$-th candidate sample. In adaptive prediction-set construction (Romano et al., 2020; Angelopoulos et al., 2020; Huang et al., 2023), the conformity score for each sample is computed via a softmax function:

$$s_i^k = \frac{\exp(-\ell(X_i, \hat{Y}_i^k))}{\sum_{j=1}^m \exp(-\ell(X_i, \hat{Y}_i^j))}, \tag{3}$$

where $-\ell$ denotes the negative log-likelihood loss. Next, the global conformity threshold $\tau$ is obtained by computing the $p$-quantile over all candidate scores:

$$\tau = Q_p \left( \left\{ s_i^j, \Big|, i = 1, \ldots, n, ; j = 1, \ldots, m \right\} \right), \tag{4}$$

$$p = \frac{\lceil (n+1)(1-\alpha) \rceil}{n}, \tag{5}$$

and $\alpha \in (0, 1)$ is the user-specified miscoverage rate. To satisfy *conditional coverage*, The prediction set for input $X_i$ is then defined as

$$C_\alpha(X_i) = \left\{ \hat{Y}_i^k, \Big|, s_i^k \geq \tau, \ k = 1, \ldots, m \right\}. \tag{6}$$

which ensures that the resulting set includes the ground truth with probability at least $1 - \alpha$. Although conformal prediction in the classical setting can provide guaranteed conditional coverage, the conformal scores assigned to candidate outputs are typically required to be *normalized*, as shown in Eq. 3, and this normalization inherently introduces a bottleneck to parallelization. Therefore, in the subsequent *Ordinal Classification*, we transform the problem into a *hypothesis testing* framework via $p$-values to avoid normalization, with a proof of its coverage guarantee provided in Appendix A.8.

**Problem Setup.** In the asynchronous test-time scaling setup, we leverage a draft model for fast sampling and delegate verification to a slower target model. Unlike classical rejection sampling (Chen et al., 2023), which approximates a target distribution with a draft distribution, we focus on accurately predicting the *rejection rate*, thereby reducing VRAM out-of-memory risk and the synchronization overhead caused by global ranking or softmax function. Given a predefined $\alpha$, we estimate a confidence level such that the ground truth $y$ falls within the prediction set $C_\alpha(Y)$ with probability at least $1 - \alpha$:

$$\mathbb{P}(y \in C_\alpha(Y)) \geq 1 - \alpha, \tag{7}$$

where $\alpha$ is conventionally interpreted as the significance level (e.g., $0.05$ corresponding to $95\%$ confidence). In this work, however, we reinterpret $\alpha$ as the *rejection rate* of the target model.

**Ordinal Classification.**   In typical inference engines (Zheng et al., 2024; Kwon et al., 2023), particularly those with asynchronous scheduling, obtaining the normalized scores for all sampling in different batches can be challenging. To avoid normalization and global ranking operations, we reformulate the task of constructing prediction sets as an *ordinal classification* (Dey et al., 2023; Xu et al., 2023), meaning that we predict the ranks of all samples. Formally, we aim to ensure:

$$\mathbb{P}(\tilde{y}^i \in C_\alpha(Y)) \geq 1 - \alpha, \quad \forall i \in \{1, \ldots, n \times m\}, \tag{8}$$

where $\mathbb{P}(\tilde{y}^i \in C_\alpha(Y))$ denotes the probability that the $i$-th candidate step $\tilde{y}^i$ lies within the prediction set $C_\alpha$, and $m$ represents the number of sampled steps. This procedure provides *marginal coverage*, meaning that the coverage guarantee holds on average over the distribution of test inputs. The stronger notion of *conditional coverage* aims to ensure

$$\mathbb{P}(\tilde{y}^i \in C_\alpha(Y) \mid X = x) \geq 1 - \alpha, \quad \forall i \in \{1, \ldots, m\}, \forall x. \tag{9}$$

That is, it provides a probabilistic guarantee for the sampled outputs corresponding to each input instance. To achieve this, our setup focuses on developing asynchronous algorithms for ranked prediction, where the construction of the prediction set ensures that its size matches the predefined budget while maintaining both marginal and conditional coverage. This approach avoids the need for normalization while addressing the challenges posed by asynchronous scheduling.

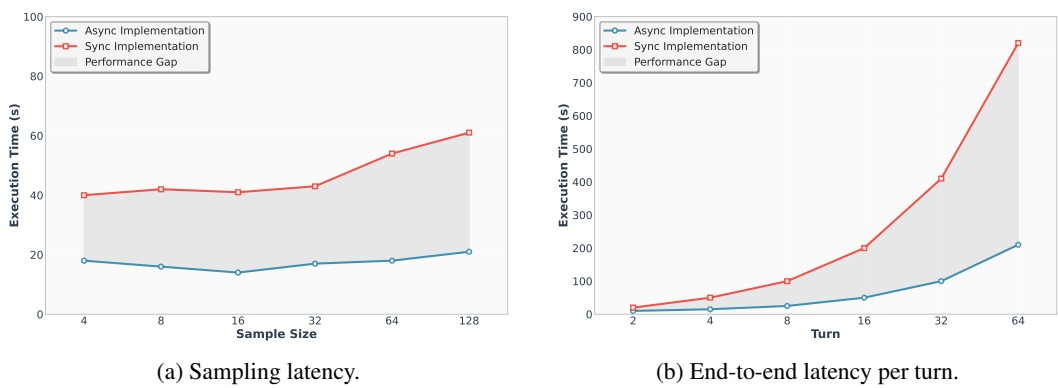

(a) Sampling latency.                                    (b) End-to-end latency per turn.

Figure 3: Execution cost comparison between synchronous and asynchronous test-time scaling.

# 3   CHALLENGE AND DESIGN

To identify performance bottlenecks in the classic setup, we introduce arithmetic intensity (Spector & Re, 2023), which measures the utilization of arithmetic units. It is defined as:

$$I = \frac{f}{b}, \tag{10}$$

where $f$ is the number of floating-point operations (FLOPs) and $b$ is the number of bytes accessed.

## 3.1   Q1: WHAT ARE THE EMERGING PERFORMANCE BOTTLENECKS?

Speculative decoding (Leviathan et al., 2023) accelerates inference by overlapping computation with memory accesses, enabling multiple draft tokens to be validated in parallel. Its main bottleneck is parallel score computation (Yin et al., 2024), making the process computation-bound.

Building upon this perspective, asynchronous scaling can be seen as an even more aggressive parallelization strategy. The target model validates far more tokens in parallel than in speculative decoding, which intensifies the prefill bottleneck and results in *total computation time far exceeding total memory access time*, as illustrated by the comparison between the green and yellow lines in Fig. 4a.

As shown in Fig. 3b, synchronization overhead grows exponentially with the number of sampling turns. In the parallel scaling setting (Fig. 3a), this overhead increases linearly with the number of concurrent samples. To this end, we observe Fig. 4a that increasing the sampling size naturally raises arithmetic intensity (with memory access time being negligible). To account for synchronization costs within arithmetic intensity, we define an *asynchronous arithmetic intensity $r$*:

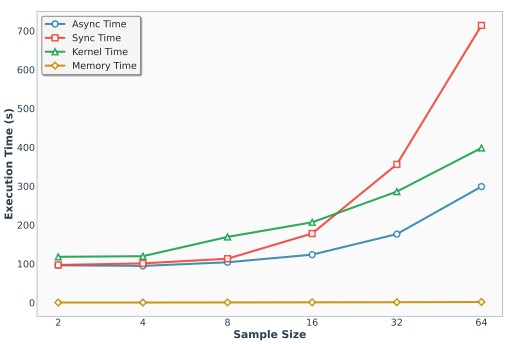

(a) Arithmetic intensity vs. sampling size.

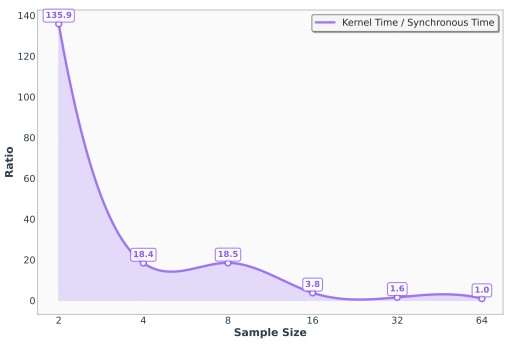

(b) Asynchronous arithmetic intensity $r$ comparison.

Figure 4: Analysis of arithmetic intensity.

$$r = \frac{T_c}{T_m + T_s} = \frac{t_c \times f}{t_m \times b + T_s} \approx \frac{T_c}{T_s}, \tag{11}$$

where $T_c$ is computation time, $T_m$ is memory access time, $t_c$ and $t_m$ are the per-unit costs of computation and memory access, respectively. It can be observed from Fig. 4b that under classic setups, $r$ decreases as the sampling size increases which indicates that synchronization overhead emerges as the *primary bottleneck*.

### 3.2 Q2: How is the prediction set constructed?

**Online Calibration.** Conformal prediction typically relies on a held-out calibration set to determine the threshold $\tau$. However, in the test-time scaling setup, held-out examples are generally unavailable. To address this limitation, we propose an *online calibration* strategy. Specifically, $m$ outputs are pre-sampled for each input in the test set, yielding $(\hat{Y}_i^1, \ldots, \hat{Y}_i^m)$. Previous efforts (Ding et al., 2023; Romano et al., 2020) impose a strict sum-to-one constraint on the conformal scores under the classification setting (where events are mutually exclusive). In contrast, we compute conformal p-values (Bates et al., 2023; Jin & Candès, 2023; Wang et al., 2024) under an ordinal classification setup, where the events are not mutually exclusive and the ordinal relationships are preserved. In this setup, we relax the strict requirement that conformity scores sum to one, and instead directly define:

$$s_\xi^k = -\ell(X_\xi, \hat{Y}_\xi^k). \tag{12}$$

This formulation is used to estimate conformal *p*-values for rejection sampling:

$$p_\xi^k = \frac{\sum_{i=1}^n \sum_{j=1}^m \mathbf{1}(s_\xi^k \leq s_i^j) + 1}{nm + 1}. \tag{13}$$

In this formulation, $s_\xi^k$ denotes the conformity score of the test-time candidate $\hat{Y}_\xi^k$, which represents the $k$-th sample of the $\xi$-th input on the test set, and $s_i^j$ are the scores from the calibration set $(X_i, \hat{Y}_i^j)$. The indicator function $\mathbf{1}(\cdot)$ returns 1 when the condition is satisfied. The $p$-value based on formula 13 guarantees *marginal coverage* at level $1 - \alpha$, which can be intuitively explained as calculating one's rank by comparing the conformity score with the score from the $p \cdot n \cdot m$ in the entire calibration set. *Conditional coverage* can be achieved by adjusting the comparison with the $p \cdot m$ calibration set from the current input sample.

A detailed proof of these two approaches is provided in Appendix A.8. The conformal $p$-value governs rejection sampling: a candidate is accepted if $p_\xi^k > \alpha$, ensuring that only high-confidence outputs are retained, thereby achieving precise budget control.

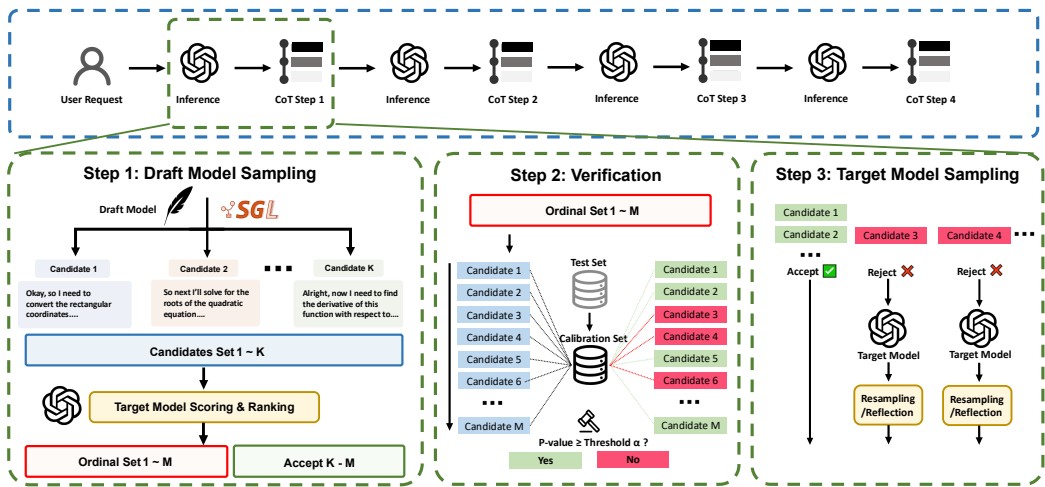

Figure 5: Asynchronous test-time scaling pipeline. The green box illustrates parallel scaling and follows the rejection sampling procedure, while the blue box illustrates sequential scaling.

**Budget Prediction.** Let $B$ denote the predefined budget (i.e., the number of candidates to reject). Given a test-time input $X_\xi$, we sample $m$ candidate CoTs $(\hat{Y}_\xi^1, \dots, \hat{Y}_\xi^m)$ in each turn and then compute their corresponding $p$-values $p_\xi^1, \dots, p_\xi^m$.

Importantly, this sampling and evaluation process is conducted *asynchronously*: each candidate is generated independently and evaluated for its $p$-value without requiring synchronization with other candidates. As a result, the outputs implicitly exhibit a descending order:

$$p_\xi^1 \geq p_\xi^2 \geq \dots \geq p_\xi^m. \tag{14}$$

The ordered set can be partitioned using a threshold to construct the *prediction set*, by directly comparing each candidate's $p$-value with the miscoverage threshold $\alpha$. Specifically, the prediction set includes all candidates whose $p$-values satisfy:

$$C_\alpha(Y_\xi) = \left\{ \hat{Y}_\xi^k : k \in \{1, \dots, m\}, p_\xi^k > \alpha \right\}. \tag{15}$$

This formulation ensures that the selected candidates meet the coverage rate. Equivalently, this can be interpreted as rejecting the top-$B$ candidate sampling.

### 3.3 How to perform rejection sampling via conformal prediction?

We adopt a three-stage sampling pipeline, illustrated in Fig. 5, to realize rejection sampling with a target rejection rate $\alpha$.

**Draft Model Sampling.** Given input tokens $x_{1:N-1}$, the draft model proposes $m$ candidate continuations of length $K_d$ in each turn, denoted $\tilde{y}_{N:N+K_d-1}^j$, by sampling from the draft model $q_d$:

$$\tilde{y}_{N:N+K_d-1}^j \sim q_d(\cdot \mid x_{1:N-1}), \quad j = 1, \dots, m, \tag{16}$$

**Verification.** For each candidate sampling $\tilde{y}_{N:N+K_d-1}^j$, we score it under the target model $q_t$ by computing the logits $q_t(\tilde{y}_{N:N+K_d-1}^j \mid x_{1:N-1})$ and converting to a conformity score. Using the calibration set, we compute a $p$-value for each candidate and reject those inside $C_\alpha$; otherwise accept.

**Target Model Sampling.** Although classical rejection sampling discards the rejected samples from the draft model and resamples entirely from the target model, to save the token budget we proceed as follows. In each turn, the per-turn target-side token budget is $K_t$: for each candidate in $C_\alpha$, we let the target model $q_t$ continue generation using that candidate from $q_d$ as a prefix for up to

Table 1: Comparison when draft and target models are from different families under marginal (Mar Cov.) and conditional (Con Cov.) coverage. $S_{\text{Mar}}$, $S_{\text{Con}}$ = end-to-end speedup (×) under marginal/conditional coverage (larger is faster); Values in red indicate lossless acceleration. *Gray rows indicate draft models (non-reasoning).*

| Dataset | Draft Model (DM) | Mar Cov. | Con Cov. | DM Baseline | TM Baseline | $S_{\text{Mar}}$ (×) | $S_{\text{Con}}$ (×) |
|---|---|---|---|---|---|---|---|
| | *QwQ-32B (RL-tuned reasoning model) as Target Model* | | | | | | |
| MATH100 | DeepSeek-R1-Distill-Qwen-1.5B | 87.0 | 94.0 | 83.0 | 96.0 | 1.8 | 1.2 |
| | Qwen2.5-7B-Instruction | 86.0 | 96.0 | 84.0 | 96.0 | 7.2 | 5.4 |
| | DeepSeek-R1-Distill-Llama-8B | 96.0 | 96.0 | 85.0 | 96.0 | 1.4 | 1.4 |
| | Llama-3.1-8B-Instruct | 87.0 | 95.0 | 75.0 | 96.0 | 2.1 | 2.2 |
| AIME24 | DeepSeek-R1-Distill-Qwen-1.5B | 86.7 | 66.7 | 60.0 | 86.7 | 4.0 | 2.6 |
| | Qwen2.5-7B-Instruction | 46.7 | 33.3 | 33.3 | 86.7 | 5.7 | 10.1 |
| | DeepSeek Llama-3.1-8B-Instruct | 80.0 | 80.0 | 80.0 | 86.7 | 2.7 | 2.3 |
| | Llama-3.1-8B-Instruct | 33.3 | 40.0 | 13.3 | 86.7 | 4.5 | 2.8 |
| AIME25 | DeepSeek-R1-Distill-Qwen-1.5B | 53.3 | 46.7 | 40.0 | 73.3 | 2.0 | 1.2 |
| | Qwen2.5-7B-Instruction | 33.3 | 40.0 | 26.7 | 73.3 | 14.5 | 12.8 |
| | DeepSeek Llama-3.1-8B-Instruct | 66.7 | 60.0 | 46.7 | 73.3 | 2.1 | 1.8 |
| | Llama-3.1-8B-Instruct | 26.7 | 33.3 | 20.0 | 73.3 | 6.7 | 2.3 |
| AMC23 | DeepSeek-R1-Distill-Qwen-1.5B | 88.0 | 90.0 | 74.0 | 94.0 | 1.0 | 1.2 |
| | Qwen2.5-7B-Instruction | 76.0 | 72.0 | 68.0 | 94.0 | 10.4 | 8.2 |
| | DeepSeek Llama-3.1-8B-Instruct | 92.0 | 94.0 | 80.0 | 94.0 | 1.5 | 1.1 |
| | Llama-3.1-8B-Instruct | 68.0 | 68.0 | 44.0 | 94.0 | 3.6 | 1.7 |
| | *s1.1-32B (SFT-tuned reasoning model) as Target Model* | | | | | | |
| MATH100 | DeepSeek-R1-Distill-Qwen-1.5B | 86.0 | 95.0 | 83.0 | 96.0 | 0.7 | 0.9 |
| | Qwen2.5-7B-Instruction | 89.0 | 96.0 | 84.0 | 96.0 | 2.9 | 2.6 |
| | DeepSeek Llama-3.1-8B-Instruct | 88.0 | 95.0 | 85.0 | 96.0 | 0.8 | 0.8 |
| | Llama-3.1-8B-Instruct | 79.0 | 85.0 | 75.0 | 96.0 | 1.3 | 1.6 |
| AIME24 | DeepSeek-R1-Distill-Qwen-1.5B | 73.3 | 80.0 | 60.0 | 86.7 | 4.4 | 2.2 |
| | Qwen2.5-7B-Instruction | 40.0 | 33.3 | 33.3 | 86.7 | 22.2 | 13.5 |
| | DeepSeek Llama-3.1-8B-Instruct | 73.3 | 73.3 | 80.0 | 86.7 | 2.4 | 3.1 |
| | Llama-3.1-8B-Instruct | 26.7 | 40.0 | 13.3 | 86.7 | 5.6 | 3.4 |
| AIME25 | DeepSeek-R1-Distill-Qwen-1.5B | 60.0 | 60.0 | 40.0 | 66.7 | 2.9 | 2.0 |
| | Qwen2.5-7B-Instruction | 33.3 | 40.0 | 26.7 | 66.7 | 18.5 | 11.9 |
| | DeepSeek Llama-3.1-8B-Instruct | 66.7 | 60.0 | 46.7 | 66.7 | 2.3 | 1.7 |
| | Llama-3.1-8B-Instruct | 26.7 | 20.0 | 20.0 | 66.7 | 5.6 | 4.2 |
| AMC23 | DeepSeek-R1-Distill-Qwen-1.5B | 86.0 | 86.0 | 74.0 | 96.0 | 1.4 | 1.0 |
| | Qwen2.5-7B-Instruction | 74.0 | 78.0 | 68.0 | 96.0 | 14.5 | 11.4 |
| | DeepSeek Llama-3.1-8B-Instruct | 92.0 | 96.0 | 80.0 | 96.0 | 1.6 | 1.6 |
| | Llama-3.1-8B-Instruct | 54.0 | 64.0 | 44.0 | 96.0 | 5.0 | 3.6 |

$K_t$ tokens, stopping earlier if an end token is encountered. In Appendix A.5, we provide a comparison between continuing sampling and resampling with $q_t$ for large-scale scaling performance.

**Termination.** We iterate the above rejection sampling until a final answer is detected, the maximum number of turns is reached, or the overall token limit is exceeded. As highlighted in Fig. 5, increasing the number of turns enables sequential test-time scaling (blue box), while increasing the number of candidates per turn enables parallel test-time scaling (green box).

## 4 EXPERIMENT

We provide the hyperparameters and other task details used in our experiments in Appendix 4 and evaluate performance under two settings: *marginal coverage* (Mar Acc.), which measures whether the budget of prediction set align on average across test inputs, and *conditional coverage* (Con Acc.), which imposes a stricter requirement that the guarantee holds for each individual input instance.

Table 2: Comparison within the same model family. $T_{\text{Th}}^m/T_{\text{Re}}^m$ and $T_{\text{Th}}^c/T_{\text{Re}}^c$ denote token-consumption ratios ($\times$) under marginal/conditional coverage relative to SpecThink (Th) and SpecReason (Re). Values in red indicate lossless acceleration (accuracy $\geq$ TM baseline). *Gray rows : Skywork-OR1 draft model.*

| Dataset | Draft / Target Model | Accuracy | | | | Token Consumption ($\times$) | | | |
| | | Mar. | Cond. | SpecTh. | SpecRe. | $T_{\text{Th}}^m$ | $T_{\text{Re}}^m$ | $T_{\text{Th}}^c$ | $T_{\text{Re}}^c$ |
|---|---|---|---|---|---|---|---|---|---|
| MATH100 | Qwen2.5-7B/32B-Instruct | **86.0** | 81.0 | 79.0 | 73.7 | 0.60 | 0.66 | 0.68 | 0.72 |
| | s1.1-7B/32B | **88.0** | 87.0 | 85.0 | 73.7 | 0.50 | 0.54 | 0.57 | 0.44 |
| | DeepSeek-R1-Distill-Qwen-1.5B/32B | **88.0** | 87.0 | 84.0 | 76.7 | 0.48 | 0.52 | 0.53 | 0.39 |
| | Skywork-OR1-7B/32B | 88.0 | **89.0** | 70.0 | 75.8 | **0.42** | **0.37** | **0.28** | **0.31** |
| AIME24 | Qwen2.5-7B/32B-Instruct | 33.3 | **40.0** | 33.3 | 33.3 | 0.61 | 0.67 | 0.69 | 0.73 |
| | s1.1-7B/32B | 66.7 | **73.3** | 40.0 | 26.7 | 0.47 | 0.52 | 0.62 | 0.50 |
| | DeepSeek-R1-Distill-Qwen-1.5B/32B | **86.7** | 80.0 | 66.7 | 66.7 | 0.46 | 0.50 | 0.56 | 0.42 |
| | Skywork-OR1-7B/32B | **86.7** | 80.0 | 60.0 | 73.3 | **0.33** | **0.27** | **0.19** | **0.24** |
| AIME25 | Qwen2.5-7B/32B-Instruct | **40.0** | 33.3 | 26.7 | **40.0** | 0.62 | 0.68 | 0.70 | 0.74 |
| | s1.1-7B/32B | **53.3** | **53.3** | 33.3 | 40.0 | 0.49 | 0.53 | 0.64 | 0.52 |
| | DeepSeek-R1-Distill-Qwen-1.5B/32B | **60.0** | 53.3 | 46.7 | 35.7 | 0.47 | 0.50 | 0.55 | 0.43 |
| | Skywork-OR1-7B/32B | **60.0** | 53.3 | 40.0 | 53.3 | **0.41** | **0.36** | **0.29** | **0.22** |
| AMC23 | Qwen2.5-7B/32B-Instruct | 72.0 | 70.0 | **74.0** | 72.0 | 0.63 | 0.69 | 0.71 | 0.75 |
| | s1.1-7B/32B | **82.0** | 78.0 | 76.0 | 78.0 | 0.48 | 0.52 | 0.62 | 0.48 |
| | DeepSeek-R1-Distill-Qwen-1.5B/32B | **92.0** | 88.0 | 82.0 | 80.0 | 0.46 | 0.50 | 0.57 | 0.44 |
| | Skywork-OR1-7B/32B | **96.0** | **94.0** | 82.0 | 86.0 | **0.39** | **0.34** | **0.37** | **0.28** |

## 4.1 ASYNCHRONOUS TEST-TIME SCALING ACROSS DIFFERENT MODEL FAMILIES.

Table 1 reports the results of asynchronous test-time scaling when the draft model (DM) and target model (TM) come from different families. We have the following **key takeaways**: i) ATTS can match the performance of the target model itself. This approach effectively reduces computational overhead while maintaining high-quality outputs, up to *22.22x* acceleration. ii) The most challenging datasets, AIME24/25, show strong performance in *marginal coverage* setup, while the other two datasets (MATH100 and AMC23) demonstrate superior results in *conditional coverage* setup. This highlights that while marginal coverage allocates more computational resources to the most difficult parts of the tasks effectively, conditional coverage ensures more reliable results at the individual input level, especially in simpler tasks, ensuring that each question is answered correctly. iii) It shows that while reasoning models consume more tokens during inference, using a reasoning model as the draft model provides better scaling performance than a non-reasoning model, though the *non-reasoning model* offers the highest acceleration. iv) When the average length of the reasoning chain output by the draft model exceeds that of the target model, the acceleration ratio is typically less than 1 on simpler datasets such as MATH and AMC23.

## 4.2 PERFORMANCE OF BUDGET PREDICTION

In this section, we evaluate the accuracy of budget prediction under marginal and conditional coverage settings. This shows how well our method controls target model interventions in rejection sampling, reflecting the accuracy of conformal prediction in estimating the rejection rate.

**Marginal Coverage.** In Figure 6b, we report the accuracy of the target-model intervention rate under marginal coverage, where the rejection rate is predicted at the dataset level. Budget-prediction accuracy across the full dataset is high, especially with the 64-sample configuration, whose absolute error stays within 5%. With $K_d = 500$ tokens for calibration and $K_d = 500$ for sampling, the error remains within 2%. This directly highlights the importance of constructing a diverse calibration set for maintaining high prediction accuracy.

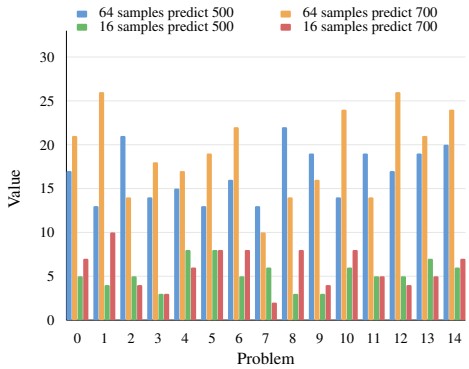
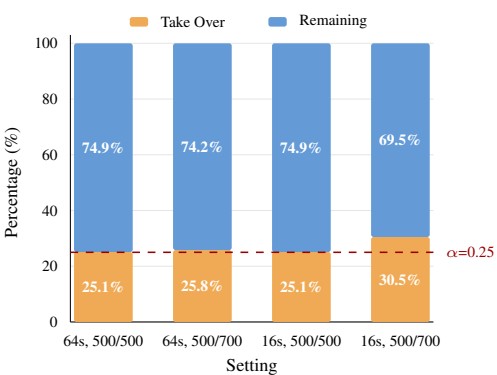

(a) Left: Conditional Coverage.

(b) Right: Marginal Coverage.

Figure 6: Budget prediction accuracy with a rejection rate $\alpha = 0.25$. (a) Per-batch conditional coverage: each bar represents the prediction error per problem under different settings ("*K samples predict L*" denotes $K$ parallel samples with calibration budget 500 and sampling budget $L$). (b) Dataset-level marginal coverage: *Take Over* denotes the fraction of samples handled by the target model, and *Remaining* denotes those kept from the draft model.

**Conditional Coverage.** In practice, we require precise *per-batch* budget control, rather than a single aggregate budget over the entire test set. Due to the limited capacity of the target model's server, it cannot process all requests concurrently. As a result, inference is performed in batches, with the token budget enforced for each batch to meet the load constraint. In Figure 6a, we report the accuracy of the target model intervention rate under conditional coverage. Under online calibration with a rejection rate of 25%, when $K_d = 500$ in both the calibration and sampling stages, the 16-sample and 64-sample settings achieve similar accuracy. However, when the calibration stage uses $K_d = 500$ but the sampling stage uses $K_d = 700$, the 64-sample setting attains significantly higher budget-prediction accuracy. This indicates that increasing the number of parallel samples can improve budget prediction accuracy when the sampling token budget differs from the calibration token budget (i.e., under a calibration–sampling token budget mismatch).

### 4.3 ASYNCHRONOUS TEST-TIME SCALING WITHIN THE SAME MODEL FAMILIES

In this section, we examine the performance across models within the same family, including both reasoning and non-reasoning models in Table 2. In this setting, since the target model and draft models share the same vocabulary, we can compare against baselines that are only applicable to models within the same family, such as Speculative Thinking (Yang et al., 2025), denoted as SPEC-THINK. Our findings are as follows: i): When the draft and target models belong to the same family, in most cases, the best performance is achieved under the setting of marginal coverage, even on simpler datasets like MATH and AMC23. ii): Across datasets, *DeepSeek* and *Skywork* show the strongest gains on challenging benchmarks (AIME, AMC), while *Qwen2.5* performs competitively on MATH100 but lags significantly on harder tasks. iii): Moreover, *s1.1* achieves moderate improvements, usually surpassing Qwen but not reaching the level of Skywork or DeepSeek. iv): Finally, *SpecReason* and *SpecThink* generally underperform compared with ATTS and consume more tokens, especially when the draft model is a reasoning model or on the more challenging AIME dataset, suggesting that their effectiveness remains limited on more complex reasoning tasks.

### 4.4 ANALYSIS OF TOKEN BUDGET AND LATENCY

**Latency and Throughput.** In this part, we analyze the trade-offs between latency and throughput under different sampling and token budget settings. i): As shown in Figure 7a, the latency of SpecReason and SpecThink consistently increases with the number of samples, highlighting the cost of scaling up sampling. In contrast, our method significantly reduces the sampling latency and achieves the lowest inference latency under the condition coverage setting. ii): The time overhead of *online calibration* is nearly negligible, particularly at larger sample sizes. iii): Meanwhile, Figure 7b illustrates how throughput varies with the per-sample token budget for methods such as *SpecReason*

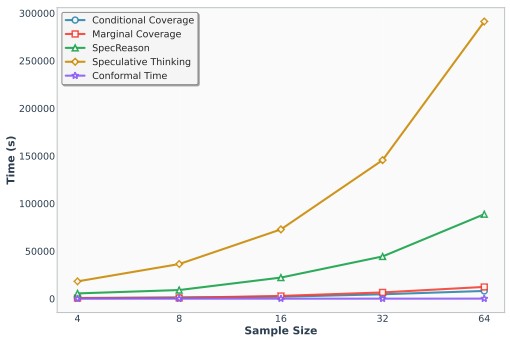
(a) Latency increases with the number of samples.

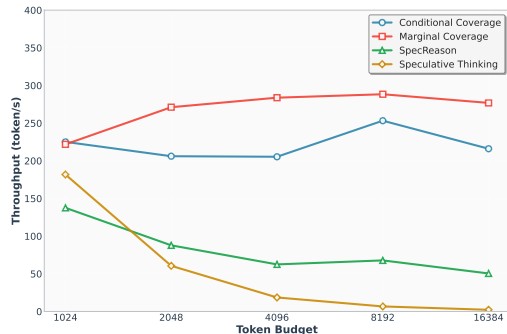
(b) Throughput variation under the 16-sample setting.

Figure 7: Analysis of latency and throughput trade-offs under different sampling and token budget.

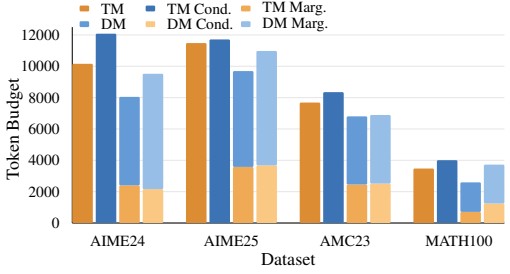
(a) Token consumption under 16-sample setting. TM/DM = Target/Draft Model; Cond./Marg. = Conditional/Marginal Coverage.

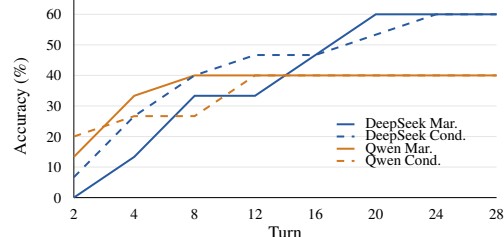
(b) Multi-turn evaluation on AIME25. Solid/dashed lines denote Mar./Cond. coverage. DeepSeek: 1.5B/S1.1-32B; Qwen: 7B/S1.1-32B.

Figure 8: (a) Token consumption under the 16-sample setting across different datasets. (b) Multi-turn evaluation results on AIME25 with increasing turns.

and *SpecThink*, revealing diminishing returns as the token budget becomes large. Our method is able to maintain high throughput even under very large budgets, especially in the marginal coverage setting. Overall, these results (Figure 7) provide insights into the balance between efficiency and performance when designing inference strategies. Under the setting of 16 samples and no limit on the maximum sequence scaling turns per sample, we achieved a **56.7x** speedup in inference and a **4.14x** throughput improvement compared to the baseline.

**Token Consumption.** Figure 8a presents the token consumption under the 16-sample setting. This includes sampling performed solely by the target model or the draft model as baselines, as well as asynchronous sampling under both condition coverage and marginal coverage settings. Compared with the two baselines, our method can significantly reduce token consumption, especially under the condition coverage setting, as it enables budget prediction at the instance level.

## 4.5 MULTI-TURN EVALUATION RESULTS

In this section, we evaluate the results on the AIME25 dataset under settings with more turns. Unlike the previous setting, we reduce the token budget per turn but increase the number of iterations, allowing the target model to generate more samples. As shown in Figure 8b, the accuracy increases with the number of turns. Both marginal coverage and conditional coverage eventually converge to the same value. The results show that under a fixed sampling size budget, increasing the number of turns still does not break the performance upper bound.

## 5 CONCLUSION

We presented ATTS (Asynchronous test-time scaling), a framework that addresses the core inefficiencies of test-time scaling in LLMs. By refining arithmetic intensity and introducing online calibration with a rejection sampling pipeline, ATTS effectively controls rejection rates while reducing latency and memory overhead. Experiments on multiple reasoning benchmarks confirm that ATTS achieves better efficiency and reliability than speculative baselines. This work establishes ATTS as a practical and principled approach for scalable test-time scaling, with potential extensions to dynamic adaptation and real-world deployment.

## ACKNOWLEDGMENTS

This work was supported in part by the Theme-based Research Scheme (TRS) project T45-701/22-R of the Research Grants Council of Hong Kong, and in part by the AVNET-HKU Emerging Microelectronics and Ubiquitous Systems (EMUS) Lab.

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

# A APPENDIX

## A.1 THE USE OF LARGE LANGUAGE MODELS

In accordance with the ICLR policy on the use of large language models, we hereby declarethat LLMs were employed solely to assist in improving the grammar and enhancing the expressionof this paper. The original research idea, methodological development, and overall structure andcontent of the manuscript were entirely conceived and written by the authors. At no stage was theuse of LLMs extended to the generation of core intellectual content, and we afirm that there hasbeen no misuse of LLMs in the preparation of this work.

## A.2 RELATED WORK

**Test-time Scaling.** Recent works explore *test-time scaling*—the idea that increasing computation during inference can be more effective than scaling model size (Snell et al., 2024; Wu et al., 2024). A common strategy is *sequential scaling*, adopted in models like OpenAI o1 (OpenAI, 2024) and DeepSeek R1 (Guo et al., 2025). Other approaches (Muennighoff et al., 2025; Yan et al., 2025) use supervised fine-tuning to match a fixed compute budget. In parallel, *parallel scaling* (Chen et al., 2025; Zeng et al., 2025; Pan et al., 2025a) improves throughput by distributing inference across replicas or devices, offering latency gains but introducing challenges in memory overhead.

**Speculative decoding** *Speculative decoding*(Li et al., 2024a; Leviathan et al., 2023; Kim et al., 2023; Chen et al., 2023) is an emerging technique for accelerating LLM inference, which is traditionally limited by slow, sequential autoregressive sampling and memory bandwidth constraints. There are three main strategies for sampling draft tokens: *token-level sampling*(Leviathan et al., 2023; Kim et al., 2023; Chen et al., 2023), where the large model directly verifies the token outputs of the draft model; *feature-level sampling* (Cai et al., 2024; Li et al., 2024a;b), which verifies generation paths using intermediate representations; and *step-level sampling* (Pan et al., 2025b; Yang et al., 2025), which operates at a coarser granularity by validating multiple tokens or computation steps together to improve throughput.

**Asynchronous Tool Calling.** The synchronization issue in batch inference with tool calls (Zhu et al., 2025) is a known obstacle to efficient reasoning. However, it remains underexplored in the context of speculative decoding—particularly when large model inference is treated as a form of tool call. In asynchronous scheduling, controlling the frequency of large model intervention is challenging due to synchronization overhead. Recent approaches (Ginart et al., 2024; Gonzalez-Pumariega et al., 2025) employ event-driven finite-state machine architectures to manage asynchronous tool calls more flexibly and efficiently.

**Conformal Prediction.** To avoid synchronization and to accurately predict the request budget in the scaling process, we introduce conformal prediction (Derhacobian et al.; Angelopoulos et al., 2020; Huang et al., 2023) to provide a theoretical guarantee for the budget of times our target model intervenes. The prediction set is then used to ensure that the large model's interventions remain consistent with the desired coverage and reliability, aligning with the validation process. However, these methods all require the model to perform a complete softmax operation (which requires synchronization), and this becomes challenging in modern inference engines with asynchronous scheduling mechanisms, thus conflicting with these methods. Some online conformal prediction algorithms (Areces et al.; Bhatnagar et al., 2023) attempt to ensure the coverage of future data in the context of online learning.

## A.3 OLYMPIADBENCH

We evaluate the results on the more challenging OlympiadBench (He et al., 2024) dataset as evidence of the robustness of our method. Under two of our settings (results shown in bold), we were even able to surpass the performance of the original target model's sampling. Under conditional coverage, we achieved a more efficient allocation of computational resources compared to marginal coverage, resulting in improved test performance. Table 3 shows the results on the OlympiadBench dataset.

Table 3: Results on the OlympiadBench benchmark. Each setting uses 16 samples and 15 turns, with $\alpha = 0.4$, 500-token budgets, and temperature 0.8.

| Draft / Target Model | Draft Model | Target Model | Marginal Cov. | Conditional Cov. |
|---|---|---|---|---|
| DeepSeek-R1-Distill-Llama-8B / S1.1-32B | 28 | 48 | 44 | 40 |
| DeepSeek-R1-Distill-Qwen-1.5B / S1.1-32B | 26 | 48 | 38 | 38 |
| Llama-3.1-8B-Instruct / S1.1-32B | 26 | 48 | **44** | **50** |
| Qwen2.5-7B-Instruct / S1.1-32B | 32 | 48 | 48 | 48 |
| DeepSeek-R1-Distill-Llama-8B / QwQ-32B | 28 | 46 | 38 | 40 |
| DeepSeek-R1-Distill-Qwen-1.5B / QwQ-32B | 26 | 46 | 38 | 40 |
| Llama-3.1-8B-Instruct / QwQ-32B | 26 | 46 | **48** | **48** |
| Qwen2.5-7B-Instruct / QwQ-32B | 32 | 46 | 50 | 46 |

### A.4 EXPERIMENTAL SETUP

We evaluate a diverse set of draft models, including DeepSeek-R1-Distill-Qwen-1.5B (Guo et al., 2025), DeepSeek-R1-Distill-Llama-8B (Guo et al., 2025), Qwen2.5-7B-Instruct (Team, 2024), Llama-3.1-8B (Dubey et al., 2024), s1.1-7B (Muennighoff et al., 2025), and Skywork-OR1-7B (He et al., 2025). Each draft model is paired with one of the large target models: QwQ-32B (Team, 2025), s1.1-32B (Muennighoff et al., 2025), Qwen2.5-32B-Instruct (Team, 2024), DeepSeek-R1-Distill-Qwen-32B (Guo et al., 2025), or Skywork-OR1-32B (He et al., 2025). We use SpecReason (Pan et al., 2025b) and Speculative Thinking (Yang et al., 2025) as baselines for speculative decoding, with a maximum length of 8192. Both of them focus on acceleration under serial scaling.

Our evaluation covers four reasoning benchmarks. We use 100 randomly sampled problems from MATH (Hendrycks et al., 2021) (denoted as MATH100) for grade-school arithmetic word problems, AIME24 (Art of Problem Solving) and AIME25 (OpenCompass) for high-school competition-level mathematics, and the first 50 problems from AMC23 (math-ai) for the American Mathematics Competitions. These datasets require multi-step reasoning and are particularly suitable for testing the effectiveness of asynchronous sampling with rejection. To ensure consistency, we set a token budget of 8192 across all settings and adopt deterministic decoding with temperature set to zero. We set the maximum number of turns to 10.

Similar to prior work (Yue et al., 2025), the best@16 metric we calculate is intended to measure the upper bound of performance for both the method and the baseline model. Unless otherwise specified, the miscoverage parameter is set to $\alpha = 0.4$, ensuring that the prediction sets are constructed with statistical guarantees. We use SGLang (Zheng et al., 2024) version 0.4.3.post4 as the inference engine. The sampling temperature is set to 0.8. We set the target model's per-turn token budget to $K_t = 500$ and the draft model's per-turn token budget to $K_d = 500$.

### A.5 LARGE-SCALE ASYNCHRONOUS TEST-SCALING

In this section, we set the maximum number of test-time scaling turns of the model to 20, and then gradually increase the number of samples per turn up to 128. We conduct comparative experiments under editing coverage, conditional coverage, as well as under the standard rejection sampling and our designed rejection sampling settings. We set the draft model to DeepSeek-R1-Distill-Qwen-1.5B, and the target model to DeepSeek-R1-Distill-Llama-70B.

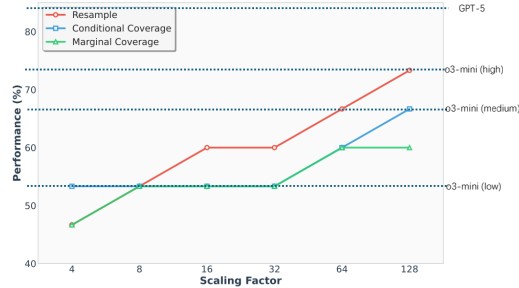

Figure 9: Accuracy improvement with increasing sample size on AIME 2025.

According to Figure 9, we can observe that as the number of samples increases, the performance of our model gradually improves. With 8 samples, our model reaches the performance of o3-mini(low); with 64 samples, it reaches the performance of o3-mini(medium); and with 128 samples, it reaches the performance of o3-mini(high), which are closed-source reasoning models. Due to

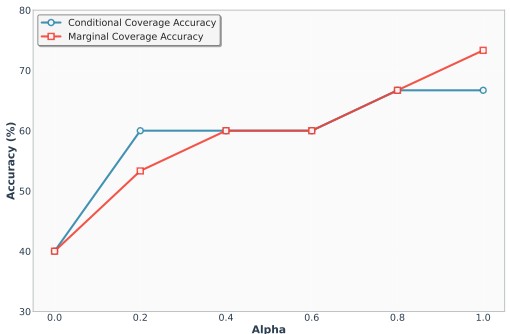 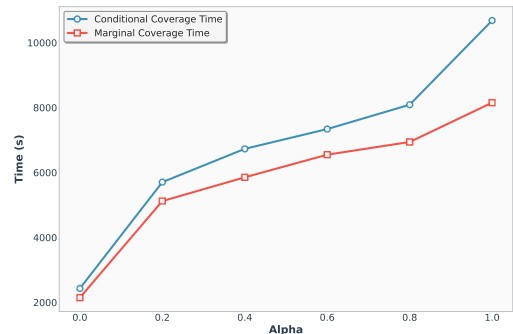

(a) The change in final accuracy as $\alpha$ increases.     (b) The increase in total time overhead as $\alpha$ grows.

Figure 10: Ablation study on the hyperparameter $\alpha$ using the DeepSeek-Qwen -1.5B and QwQ-32B models on the AIME2025 dataset.

limited computational resources, we did not conduct experiments with larger-scale sampling, which results in still falling short of GPT-5 performance.

**Continue Sampling.** In our experimental setup, we did not strictly adhere to the standard rejection sampling procedure (i.e., discarding the samples generated by the draft model and resampling with the target model) when performing scaling. Instead, under the continue sampling setting, if a sample produced by the draft model is included in the prediction set of the current turn, the target model subsequently continues the sampling in the following turn conditioned on this sampled result. According to Figure 9, both our conditional coverage and editing coverage adopt the continue-sampling scheme. The conditional coverage demonstrates relatively high sampling efficiency, reaching the level of o3-mini-medium under the 128-sample setting.

**Resampling.** We also conducted experiments that scale the number of samples under the standard rejection sampling setting. In this setting, at each scaling turn, if the draft model's sample is included in the prediction set for the current turn, then within the same turn the target model draws a prediction set whose size matches that of the current prediction set. As indicated by the red curve in Figure 9, this scheme exhibits substantially higher sampling efficiency than the alternatives; however, for the same nominal number of samples it consumes more tokens (since part of the draft model's tokens are discarded). Under the 128-sample setting, it achieves performance comparable to o3-mini-high.

### A.6 ABLATION STUDY

In this section, we present the ablation experiments on the hyperparameter $\alpha$. As shown in Figure 10, as the hyperparameter $\alpha$ increases, the overall accuracy and time overhead of the system both rise. This reflects that the intervention of the target model increases both accuracy and time overhead. However, we can find a balance between accuracy and time overhead, such as when $\alpha$ is 0.2, where the condition coverage setting achieves 60% accuracy with relatively low time overhead. When $\alpha$ is 0.4, both the marginal coverage and condition coverage settings reach 60% accuracy, making it a more robust hyperparameter setting.

### A.7 CONTROLLING THE REJECTION RATE IN MULTI-TURN INTERACTIONS

In this section, we study the variation of model perplexity with sequential scaling and the change in the number of take-overs by the large model during rejection sampling. This directly reflects whether the conformal prediction algorithm we adopt can control the intervention rate of the target model within an acceptable range. i): According to Figure 11a, each generation by the draft model leads to an increase in the overall PPL, while the intervention of the target model effectively mitigates this trend. As the number of turns increases, the PPL of the scaling process can be gradually reduced, which indirectly results in a decrease in the rejection rate. ii): Meanwhile, in Figure 11b, we directly visualize the change in the number of take-overs by the target model as the number of interaction turns increases. We observe that with sequential scaling, the target model overall maintains a rejection rate around a fixed level, which gradually decreases and eventually approaches zero. iii): The target model continuously adjusts the convergence behavior of the draft model's perplexity during sequential scaling.

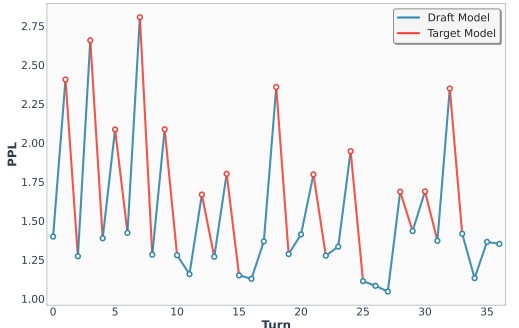 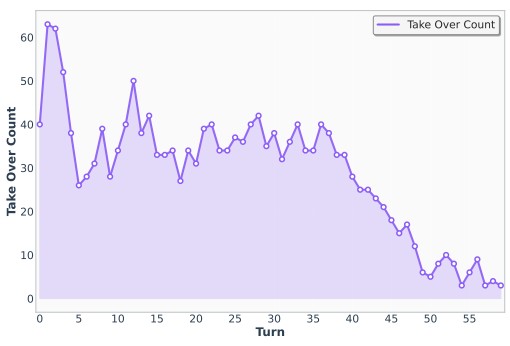

(a) PPL variation with increasing turns.     (b) Take-over counts across multiple turns.

Figure 11: Analysis of model behavior in multi-turn interactions. The left subfigure shows how the Perplexity (PPL) of draft and target models evolves as the number of turns increases, while the right subfigure presents the take-over counts across turns.

## A.8 DEFINITION AND PROOF

**Proposition 1.** *Suppose that $\{(X_i, Y_i)\}_{i=1}^n$ are exchangeable random variables from the test dataset, and $\xi \sim Uniform\{1, 2, \ldots, n\}$ represents randomly sampling one data point from the test dataset, where $k$ denotes the $k$-th sample of that data point, then the marginal conformal p-values defined as,*

$$p_\xi^k = \frac{\sum_{i=1}^n \sum_{j=1}^m \mathbf{1}(s_\xi^k \leq s_i^j) + 1}{nm + 1} \tag{17}$$

*is valid in the sense that for the miscoverage rate $\alpha \in (0, 1)$, we have*

$$\mathbb{P}(p_\xi^k \leq \alpha) \leq \alpha. \tag{18}$$

*Moreover, if the conformity scores $\{s_i^j\}_{i=1,j=1}^{n,m}$ are distinct surely, we have,*

$$p_\xi^k \sim U\left\{\frac{1}{nm+1}, \frac{2}{nm+1}, \ldots, 1\right\}. \tag{19}$$

*Proof of Proposition 1.* Suppose, for any given values of conformity scores, $v_1, \ldots, v_{nm+1}$, they can be rearranged as $\tilde{v}_1 < \cdots < \tilde{v}_\ell$ with repetitions $n_i$ of $\tilde{v}_i$ such that $\sum_{i=1}^\ell n_i = nm + 1$. Let $E_v$ denote the event of $\{s_1^1, s_1^2, \ldots, s_n^m, s_\xi^k\} = \{v_1, \ldots, v_{nm+1}\}$.

Then, under $E_v$, for $i = 1, \ldots, \ell$, we have

$$\mathbb{P}(s_\xi^k = \tilde{v}_i \mid E_v) = \frac{n_i}{nm + 1}, \tag{20}$$

due to the exchangeability of conformity scores.

We also note that under $E_v$ and $s_\xi^k = \tilde{v}_i$ we have from Equation equation 17,

$$p_\xi^k = \frac{\sum_{l=1}^i n_l}{nm + 1}. \tag{21}$$

Then, for any $\alpha \in [0, 1]$ and $i = 1, \ldots, \ell$, we have

$$\mathbb{P}(p_\xi^k \leq \alpha \mid E_v, s_\xi^k = \tilde{v}_i) = \begin{cases} 0 & \text{if } \alpha < \frac{\sum_{l=1}^i n_l}{nm+1}, \\ 1 & \text{otherwise.} \end{cases} \tag{22}$$

Thus, for any $i = 1, \ldots, \ell$ and $\frac{\sum_{l=1}^{i-1} n_l}{nm+1} \leq \alpha < \frac{\sum_{l=1}^{i} n_l}{nm+1}$, we have

$$\mathbb{P}(p_\xi^k \leq \alpha \mid E_v) = \sum_{l=1}^{\ell} \mathbb{P}(p_\xi^k \leq \alpha \mid E_v, s_\xi^k = \tilde{v}_l) \cdot \mathbb{P}(s_\xi^k = \tilde{v}_l \mid E_v) \tag{23}$$

$$= \frac{\sum_{l=1}^{i-1} n_l}{nm+1} \leq \alpha. \tag{24}$$

By taking the expectation over the above inequality, it follows that the conformal $p$-value $p_\xi^k$ is marginally valid.

Specifically, if conformity scores $\{s_i^j\}_{i=1,j=1}^{n,m} \cup \{s_\xi^k\}$ are distinct surely, then $\ell = nm+1$ and $n_i = 1$ for $i = 1, \ldots, nm+1$. Thus,

$$\mathbb{P}(p_\xi^k \leq \alpha \mid E_v) = \frac{i-1}{nm+1}, \quad \text{if } \frac{i-1}{nm+1} \leq \alpha < \frac{i}{nm+1}, \tag{25}$$

that is, $p_\xi^k \mid E_v \sim U\left\{\frac{1}{nm+1}, \frac{2}{nm+1}, \ldots, 1\right\}$. This completes the proof. Next, we address the theoretical part that guarantees conditional coverage. $\qquad\square$

**Proposition 2.** *Suppose that $\{(X_i, Y_i)\}_{i=1}^n$ are exchangeable random variables from the test dataset, then for any sample $y \in \mathcal{Y}$, given a conditioning set $\mathcal{I}_y \subseteq \{0, \ldots, m-1\}$ for each $X_i$ constructed based on the specific sample $y$ for a test input $x$ and $Y_\xi = y$, the corresponding conditional conformal p-value as defined in Equation equation 17, is conditionally valid in the sense that for any $\alpha \in [0, 1]$,*

$$\mathbb{P}(p_\xi^k \leq \alpha \mid \mathcal{I}_y, Y_\xi = y) \leq \alpha. \tag{26}$$

*Moreover, if $\{s_i^j\}_{j \in \mathcal{I}_y, i=1,\ldots,n}$ are distinct surely, we have that conditional on $\mathcal{I}_y$ and $Y_\xi = y$,*

$$p_\xi^k \sim U\left\{\frac{1}{m+1}, \frac{2}{m+1}, \ldots, 1\right\}, \tag{27}$$

*where $m = |\mathcal{I}_y|$ is the size of the conditioning set for output $y$.*

*Proof.* For any given sample $y \in \mathcal{Y}$, the corresponding conditional conformal $p$-value is given by

$$p_\xi^k = \frac{1}{m+1}\left(\sum_{i=1}^n \sum_{j \in \mathcal{I}_y} \mathbf{1}\{s_i^j \leq s_\xi^k\} + 1\right), \tag{28}$$

where $\mathcal{I}_y \subseteq \{0, \ldots, m-1\}$ is the conditioning set for sample $y$, $m = |\mathcal{I}_y|$, $s_i^j$ represents the conformity score for the $j$-th candidate of the $i$-th test instance, and $s_\xi^k$ is the conformity score for the $k$-th candidate of the new test instance.

Given $\mathcal{I}_y$ and $Y_\xi = y$, the conformity scores $\{s_i^j\}_{j \in \mathcal{I}_y, i=1,\ldots,n} \cup \{s_\xi^k\}$ are exchangeably distributed, which follows from the assumption that $\{(X_i, Y_i)\}_{i=1}^n$ are exchangeably distributed and the construction of candidate outputs.

Using similar arguments as in the proof of Proposition 1, for any given values of conformity scores $v_1, \ldots, v_{nm+1}$, suppose that they can be arranged as $\tilde{v}_1 < \cdots < \tilde{v}_\ell$ with repetitions $m_i$ of $\tilde{v}_i$ such that $\sum_{i=1}^{\ell} m_i = nm+1$. Let $E_v$ denote the event $\{s_i^j\}_{j \in \mathcal{I}_y, i=1,\ldots,n} \cup \{s_\xi^k\} = \{v_1, \ldots, v_{nm+1}\}$.

Then, given $E_v$, $\mathcal{I}_y$, and $Y_\xi = y$, we have

$$\mathbb{P}(s_\xi^k = \tilde{v}_i \mid E_v, \mathcal{I}_y, Y_\xi = y) = \frac{m_i}{nm+1} \tag{29}$$

for $i = 1, \ldots, \ell$, due to exchangeability of the conformity scores.

Note that given $E_v$, $\mathcal{I}_y$, $Y_\xi = y$, and $s_\xi^k = \tilde{v}_i$, we have from Equation equation 28,

$$p_\xi^k = \frac{\sum_{j=1}^{i} m_j}{nm+1}. \tag{30}$$

Thus, for any $\alpha \in [0,1]$ and $i = 1, \ldots, \ell$,

$$\mathbb{P}(p_\xi^k \leq \alpha \mid E_v, \mathcal{I}_y, Y_\xi = y, s_\xi^k = \tilde{v}_i) = \begin{cases} 0 & \text{if } \alpha < \frac{\sum_{j=1}^i m_j}{nm+1}, \\ 1 & \text{otherwise.} \end{cases} \tag{31}$$

Then, for any given $i = 1, \ldots, \ell$ and $\frac{\sum_{j=1}^{i-1} m_j}{nm+1} \leq \alpha < \frac{\sum_{j=1}^i m_j}{nm+1}$, we have

$$\mathbb{P}(p_\xi^k \leq \alpha \mid E_v, \mathcal{I}_y, Y_\xi = y) \tag{32}$$

$$= \sum_{j=1}^{\ell} \mathbb{P}(p_\xi^k \leq \alpha \mid E_v, \mathcal{I}_y, Y_\xi = y, s_\xi^k = \tilde{v}_j) \cdot \mathbb{P}(s_\xi^k = \tilde{v}_j \mid E_v, \mathcal{I}_y, Y_\xi = y) \tag{33}$$

$$= \frac{\sum_{j=1}^{i-1} m_j}{nm+1} \leq \alpha. \tag{34}$$

By taking expectation, it follows that $p_\xi^k$ is conditionally valid given $Y_\xi = y$. This completes the proof. □

**Discussion.** After proving the validity of individual conformal p-values, in order to obtain the rejection rate for the entire test set, that is, to ensure that the overall error rate is controlled when simultaneously testing K hypotheses, we propose the following Proposition.

**Proposition 3.** *Suppose that $\{(X_i, Y_i)\}_{i=1}^n$ are exchangeable random variables, and let $\xi \sim Uniform\{1, 2, \ldots, n\}$ represent a randomly selected test instance, then* A1 *based on marginal conformal p-values all provide simultaneous coverage guarantees across the entire test dataset at level $1 - \alpha$, i.e.,*

$$\mathbb{P}(\forall i \in \{1, \ldots, n\} : Y_i \in C(X_i)) \geq 1 - \alpha. \tag{35}$$

*Specifically, if the conformity scores $\{s_i^j\}_{i=1,j=1}^{n,m}$ are distinct surely, then for* A1*, we also have,*

$$\mathbb{P}(\forall i \in \{1, \ldots, n\} : Y_i \in C(X_i)) \geq 1 - \alpha + \frac{1}{(n-1)m+1}. \tag{36}$$

*Proof of Proposition 3.* Consider A1 based on marginal conformal p-values. Note that among the tested hypotheses $H_1, \ldots, H_m$, there is exactly one hypothesis $H_{Y_\xi}$ to be true. Thus, the probability that all samples are correctly covered by A1 satisfies:

$$\mathbb{P}(\forall i \in \{1, \ldots, n\} : Y_i \in C(X_i)) = \mathbb{P}(\text{accept } H_{Y_\xi}) \tag{37}$$

$$= 1 - \mathbb{P}(\text{reject } H_{Y_\xi}) \tag{38}$$

$$\geq 1 - \mathbb{P}(p_\xi^{Y_\xi} \leq \alpha) \tag{39}$$

$$\geq 1 - \alpha, \tag{40}$$

where the last inequality follows by Proposition 1.

Specifically, for A1, if the conformity scores $\{s_i^j\}_{i=1,j=1}^{n,m}$ are distinct surely, by Proposition 1, we have

$$\mathbb{P}(\forall i \in \{1, \ldots, n\} : Y_i \in C(X_i)) = \mathbb{P}(\text{accept } H_{Y_\xi}) \tag{41}$$

$$= 1 - \mathbb{P}(p_\xi^{Y_\xi} \leq \alpha) \tag{42}$$

$$\geq 1 - \left(\alpha - \frac{1}{(n-1)m+1}\right) \tag{43}$$

$$= 1 - \alpha + \frac{1}{(n-1)m+1}, \tag{44}$$

which gives the desired result. □

**Theorem 1.** *Suppose that $\{(X_i, Y_i)\}_{i=1}^n$ are exchangeable random variables, and let $\xi \sim Uniform\{1, 2, \ldots, n\}$ represent a randomly selected test instance, then the prediction set $C(X_\xi) = \left\{ \hat{Y}_\xi^k \mid p_\xi^k > \alpha \right\}$ determined by* A1 *both satisfy*

$$\mathbb{P}(Y_\xi \in C(X_\xi)) \geq 1 - \alpha. \tag{45}$$

*Proof.* Note that the prediction set is given by $C(X_\xi) = A_1 \cap A_2$. Thus, by Proposition 1,

$$\mathbb{P}(Y_\xi \in C(X_\xi)) \geq \mathbb{P}(p_\xi^{Y_\xi} > \alpha) \geq 1 - \alpha. \tag{46}$$

Similarly, its prediction set is given by

$$C(X_\xi) = \{y \in \mathcal{Y} : p_\xi^y > \alpha\}. \tag{47}$$

By Proposition 1, it is easy to check that

$$\mathbb{P}(Y_\xi \in C(X_\xi)) = \mathbb{P}(p_\xi^{Y_\xi} > \alpha) \geq 1 - \alpha. \tag{48}$$

Specifically, if the conformity scores $\{s_i^j\}_{i=1, j=1}^{n,m}$ are distinct surely, we have

$$\mathbb{P}(Y_\xi \in C(X_\xi)) = 1 - \mathbb{P}(p_\xi^{Y_\xi} \leq \alpha) \leq 1 - \alpha + \frac{1}{(n-1)m+1}. \tag{49}$$

This completes the proof. $\qquad\square$

**Discussion.** Now that we have completed the proof of marginal coverage, we proceed to prove the conditional coverage for the entire test dataset. Under the exchangeability assumption, we have:

**Proposition 4.** *Under the same exchangeability assumption as in Proposition 2,* A1 *based on conditional conformal $p$-values $p_\xi^k$ all provide conditional coverage guarantees for the entire test dataset at level $1 - \alpha$, i.e., for any $y \in \mathcal{Y}$,*

$$\mathbb{P}(\forall i \in \{1, \ldots, n\} : Y_i \in C(X_i) \mid Y_\xi = y) \geq 1 - \alpha. \tag{50}$$

*Specifically, if the conformity scores $\{s_i^j\}_{j \in \mathcal{I}_y, i=1,\ldots,n}$ are distinct surely, then for* A1 *based on $p_\xi^k$, we have that for any $y \in \mathcal{Y}$ and $\mathcal{I}_y \subseteq \{0, \ldots, m-1\}$,*

$$\mathbb{P}(\forall i \in \{1, \ldots, n\} : Y_i \in C(X_i) \mid Y_\xi = y, \mathcal{I}_y) \geq 1 - \alpha + \frac{1}{(n-1)|\mathcal{I}_y|+1}. \tag{51}$$

*Proof.* Consider Procedure 1-3 based on conditional conformal $p$-values. For any $y \in \mathcal{Y}$, given $Y_\xi = y$, the conditional coverage probability for the entire test dataset satisfies:

$$\mathbb{P}(\forall i \in \{1, \ldots, n\} : Y_i \in C(X_i) \mid Y_\xi = y) = 1 - \mathbb{P}(\text{reject } H_y \mid Y_\xi = y) \tag{52}$$
$$\geq 1 - \mathbb{P}(p_\xi^y \leq \alpha \mid Y_\xi = y) \tag{53}$$
$$\geq 1 - \alpha, \tag{54}$$

where the inequalities follow from the definitions of Procedure 1-3 and Proposition 2.

Specifically, for Procedure 3, if the conformity scores $\{s_i^j\}_{j \in \mathcal{I}_y, i=1,\ldots,n}$ are distinct surely, then by Proposition 2, the coverage probability conditional on $\mathcal{I}_y$ and $Y_\xi = y$ satisfies:

$$\mathbb{P}(\forall i \in \{1, \ldots, n\} : Y_i \in C(X_i) \mid \mathcal{I}_y, Y_\xi = y) = 1 - \mathbb{P}(\text{reject } H_y \mid \mathcal{I}_y, Y_\xi = y) \tag{55}$$
$$= 1 - \mathbb{P}(p_\xi^y \leq \alpha \mid \mathcal{I}_y, Y_\xi = y) \tag{56}$$
$$\geq 1 - \left( \alpha - \frac{1}{(n-1)|\mathcal{I}_y|+1} \right) \tag{57}$$
$$= 1 - \alpha + \frac{1}{(n-1)|\mathcal{I}_y|+1}. \tag{58}$$

This completes the proof. $\qquad\square$

**Theorem 2.** *Under the same exchangeability assumption as in Theorem 1, the prediction set $C(X_\xi \mid y)$ based on conditional conformal p-values $p_\xi^k$ satisfies*

$$\mathbb{P}(Y_\xi \in C(X_\xi \mid y) \mid Y_\xi = y) \geq 1 - \alpha \tag{59}$$

*for any $y \in \mathcal{Y}$. Specifically, for the prediction set $C(X_\xi \mid y)$ based on $p_\xi^k$, if the conformity scores $\{s_i^j\}_{j \in \mathcal{I}_y, i=1,\ldots,n}$ are distinct surely, we have*

$$\mathbb{P}(Y_\xi \in C(X_\xi \mid y) \mid Y_\xi = y, \mathcal{I}_y) \leq 1 - \alpha + \frac{1}{(n-1)|\mathcal{I}_y| + 1} \tag{60}$$

*for any $y \in \mathcal{Y}$ and $\mathcal{I}_y \subseteq \{0, \ldots, m-1\}$.*

*Proof.* By using Proposition 4 and similar arguments as in the proof of Theorem 1, the prediction sets $C(X_\xi \mid y)$ based on the conditional conformal $p$-values $p_\xi^k$ all satisfy:

$$\mathbb{P}(Y_\xi \in C(X_\xi \mid y) \mid \mathcal{I}_y, Y_\xi = y) \geq 1 - \alpha \tag{61}$$

for any $y \in \mathcal{Y}$.

Specifically, if the conformity scores $\{s_i^j\}_{j \in \mathcal{I}_y, i=1,\ldots,n}$ are distinct surely, we have:

$$\mathbb{P}(Y_\xi \in C(X_\xi \mid y) \mid \mathcal{I}_y, Y_\xi = y) = \mathbb{P}(p_\xi^y > \alpha \mid \mathcal{I}_y, Y_\xi = y) \tag{62}$$

$$\leq 1 - \alpha + \frac{1}{(n-1)|\mathcal{I}_y| + 1}, \tag{63}$$

which gives the desired result. $\square$

### A.9 PROMPT

---

**Completion Question**: Please answer the following problem using step-by-step reasoning. Please separate your reasoning steps with two newline characters (\\n \\n). Please must put your final answer within \\boxed{{}}.
Question: {question}

**Multiple Choice Question**: This is a multiple-choice question. Please answer the following problem using step-by-step reasoning. Separate each reasoning step with two newline characters (\\n \\n). You must put your final answer within \\boxed{{}}, such as \\boxed{{A}}, \\boxed{{B}}, \\boxed{{C}}, or \\boxed{{D}}. No other formats are allowed.
Question: {question}
Choices: A. {choice[1]} B. {choice[2]} C. {choice[3]} D. {choice[4]}

---

