# OpenReview forum: "ATTS: Asynchronous Test-Time Scaling via Conformal Prediction"
_ICLR.cc/2026/Conference — ICLR 2026 Poster_

### Official Review · Reviewer_ct8q · 2025-10-31

**Soundness:** 3
**Presentation:** 3
**Contribution:** 3
**Rating:** 2
**Confidence:** 4

**Summary:**

This paper proposes ATTS (Asynchronous Test-Time Scaling) to address the high latency problem in large model inference. The authors introduce the concept of asynchronous arithmetic intensity to measure the ratio between the time spent sampling candidates and the time waiting for the large model to verify them, showing that as the number of samples increases, the cost from waiting for verification becomes more significant. The method leverages ideas from conformal prediction to reduce the candidate set that requires verification by the large model, thereby lowering compute costs. The resulting ATTS framework applies conformal prediction for theoretically lossless speedup of test-time inference.

**Strengths:**

The paper provides valuable insights into how to reduce the time cost of test-time scaling (TTS), especially through the lens of asynchronous arithmetic intensity.

Effectively applies conformal prediction to minimize the workload of the large model during inference, and demonstrates both theoretical and practical speedup.

The experimental section covers several representative math reasoning tasks, and the results are fairly comprehensive for the considered setting.

**Weaknesses:**

The method is not empirically compared against other prominent recent approaches in the field, such as SpecReason and Speculative Thinking, making it difficult to objectively evaluate the unique advantages of ATTS.

Experiments suggest that when the small and large models are trained on data from different distributions, the benefits of ATTS decrease noticeably, highlighting a potential issue: the method does not fundamentally address the extra rejection cost caused by distribution mismatch between the small and large models.

Theoretical guarantees rely on key assumptions from conformal prediction (e.g., exchangeability and normalization of scores). However, the implementation omits score normalization, which may break these assumptions and cause the rejection rate control to fail, especially in multi-task or distribution-shifted settings.

**Questions:**

1.Validity of theoretical assumptions
The omission of normalization in the implementation may break the core assumptions of conformal prediction, potentially leading to excessive rejections for some tasks and too few for others in mixed or real-world scenarios. How does the method ensure fairness and validity of the rejection rate when handling multi-task or large-scale deployments?

2.Lack of comparison with strong baselines
The experimental section does not compare ATTS with other competitive methods such as SpecReason or Speculative Thinking. What are the practical or theoretical advantages of ATTS over these methods, and is there any empirical evidence of its superiority or unique use cases?

3.System/engineering considerations
The paper does not discuss the impact of modern inference engine features, such as supporting asynchronous verification requests. Can ATTS be integrated with inference backends that support asynchronous calls, and is there any experimental evidence for further gains from such integration?

---

> ### Author Response · Authors · 2025-11-13
> **Response to Reviewer ct8q Part I**
>
> Thank you for your constructive review comments, and also for acknowledging that our work provides valuable insights into how to reduce the time cost of test-time scaling.
>
> **Q1:** The method is not empirically compared against other prominent recent approaches in the field, such as SpecReason and Speculative Thinking, making it difficult to objectively evaluate the unique advantages of ATTS.
>
> **A1:**  In our submission, we have already compared our method with SpecReason and Speculative Thinking, which are respectively denoted as SpecReason and SpecThink in **Table 2 of our paper**. Since the SpecThink method requires the draft model and the target model to share the same vocabulary, for fairness, we conduct comparisons with these two baselines under the setting where all models belong to the same model family.

---

> ### Author Response · Authors · 2025-11-13
> **Response to Reviewer ct8q Part II**
>
> **Q2:** Experiments suggest that when the small and large models are trained on data from different distributions, the benefits of ATTS decrease noticeably, highlighting a potential issue: the method does not fundamentally address the extra rejection cost caused by distribution mismatch between the small and large models.
>
> **A2:** According to our understanding, what you referred to as “the small and large models being trained on data from different distributions” means that the large model is trained on long reasoning chain data (a reasoning model), while the small model is fine-tuned on instruction-following data (a non-reasoning model). Based on the experimental results in Table 1 of our paper, this combination—where there is an output distribution mismatch between the small and large models—achieves up to a 22.22× speedup on the AIME dataset, which requires complex reasoning. Therefore, the statement about an “extra rejection cost” is puzzling. Could you please elaborate on your viewpoint further in light of the results in Table 1?

---

> ### Author Response · Authors · 2025-11-13
> **Response to Reviewer ct8q Part III**
>
> The key is construction of p-values to replace probability.
>
> **Q3:** Theoretical guarantees rely on key assumptions from conformal prediction (e.g., exchangeability and normalization of scores). However, the implementation omits score normalization, which may break these assumptions and cause the rejection rate control to fail, especially in multi-task or distribution-shifted settings.
>
> **A3:** The basic assumption of conformal prediction is the exchangeability (or i.i.d., or its weakened form) of the output, which does not necessarily have to satisfy the standard form of a probability distribution. **The key is construction of p-values to replace probability.** In many papers[1, 2, 3,4], the "conformity score" and p-value mechanisms are used to generate prediction sets/intervals. These mechanisms do not explicitly require the model's output to form a standard probability distribution (where the sum of probabilities for all classes equals 1) as a premise.
>
> - In the test-time scaling setting, this assumption of exchangeability is clearly satisfied.
> - The fact that the softmax outputs sum to 1 is meant to satisfy the assumption that the probabilities of all mutually exclusive events add up to 1. Implicitly, this also means that each event (class) is independent and mutually exclusive from the others — that is, the model is effectively performing *nominal classification*.
> - In ordinal classification, the categories are not mutually exclusive events but rather monotonically ordered levels. Therefore, it is not necessary for the probabilities of all events to sum to one.
> - By using independent conformity scores, such as PPL (any non-negative real-valued function can serve as a nonconformity measure), together with a multiple testing framework, we can more naturally control the coverage rate while preserving the ordinal structure.
> - In Appendix A.7, we provide a proof showing that reformulating the classification problem as a set of ordered hypothesis tests guarantees both conditional and marginal coverage under a specified family-wise error rate(α).
>
> **Reference:**
> [1] Conformal Prediction: a Unified Review of Theory and New Challenges.     2023, Journal of the Bernoulli Society for Mathematical Statistics and Probability
> [2]  Testing for Outliers with Conformal p‑values.  2022, Annals of Statistics
> [3] Selection by Prediction with Conformal p‑values. 2023, Journal of Machine Learning Research
> [4] Conformalized Multiple Testing after Data-dependent Selection.  2024, NeurIPS

---

> ### Author Response · Authors · 2025-11-13
> **Response to Reviewer ct8q Part IV**
>
> **Q4:** Validity of theoretical assumptions The omission of normalization in the implementation may break the core assumptions of conformal prediction, potentially leading to excessive rejections for some tasks and too few for others in mixed or real-world scenarios. How does the method ensure fairness and validity of the rejection rate when handling multi-task or large-scale deployments?
>
> **A4:**  Thank you for your valuable feedback.
> - Regarding the assumption that the output probabilities must sum to 1, we have already discussed this in the previous question — that is, we use conformal p-values instead of probabilities to construct the prediction set. This approach[1,2,3,4] has been discussed in many papers.
>
> - Regarding the control of rejection rates for **handling multi-task settings**, this is a scenario we have not considered before. In fact, we can treat the task type as a form of conditional probability and use the conformal p-values proposed in our paper to ensure conditional coverage.
>
> - Regarding large-scale deployments, our paper’s precise control of the rejection rate is designed to accurately manage the computational and memory load on the server hosting the target model, preventing GPU memory overflow and solving load-balancing issues. We can precompute the maximum concurrency for the target model at a given max inference length—based on the server’s processor core count, GPU memory capacity, and the size of the deployed model (by running a profiling step)—which then determines the corresponding rejection rate.  **We conducted experiments under a large-scale sampling setup (128 samples per round). As shown in Appendix A.4 of the paper, our method can match the state-of-the-art GPT o3-mini (high), demonstrating the superiority of our approach.**
>
> **Reference:**
> [1] Conformal Prediction: a Unified Review of Theory and New Challenges.     2023, Journal of the Bernoulli Society for Mathematical Statistics and Probability
> [2]  Testing for Outliers with Conformal p‑values.  2022, Annals of Statistics
> [3] Selection by Prediction with Conformal p‑values. 2023, Journal of Machine Learning Research
> [4] Conformalized Multiple Testing after Data-dependent Selection.  2024, NeurIPS

---

> ### Author Response · Authors · 2025-11-13
> **Response to Reviewer ct8q Part V**
>
> **Q5:** Lack of comparison with strong baselines The experimental section does not compare ATTS with other competitive methods such as SpecReason or Speculative Thinking. What are the practical or theoretical advantages of ATTS over these methods, and is there any empirical evidence of its superiority or unique use cases?
>
> **Table 2. Comparison when the draft and target models are from the same family.**
> *Token consumption ratios (×) under marginal (Tᵐ) and conditional (Tᶜ) coverage, relative to SpecThink and SpecReason baselines (smaller is better).*
>
> | Dataset | Draft / Target Model | Mar Cov. | Con Cov. | SpecThink | SpecReason | Tᵐ_Think | Tᵐ_Reason | Tᶜ_Think | Tᶜ_Reason |
> |:--------:|:--------------------|:---------:|:---------:|:----------:|:-----------:|:--------:|:----------:|:---------:|:----------:|
> | **MATH100** | Qwen2.5-7B / 32B | **86.0** | 81.0 | 79.0 | 73.7 | 0.60 | 0.66 | 0.68 | 0.72 |
> |  | s1.1-7B / 32B | **88.0** | 87.0 | 85.0 | 73.7 | 0.50 | 0.54 | 0.57 | 0.44 |
> |  | DeepSeek-R1-1.5B / 32B | **88.0** | 87.0 | 84.0 | 76.7 | 0.48 | 0.52 | 0.53 | 0.39 |
> |  | Skywork-OR1-7B / 32B | 88.0 | **89.0** | 70.0 | 75.8 | **0.42** | **0.37** | **0.28** | **0.31** |
> | **AIME24** | Qwen2.5-7B / 32B | 33.3 | **40.0** | 33.3 | 33.3 | 0.61 | 0.67 | 0.69 | 0.73 |
> |  | s1.1-7B / 32B | 66.7 | **73.3** | 40.0 | 26.7 | 0.47 | 0.52 | 0.62 | 0.50 |
> |  | DeepSeek-R1-1.5B / 32B | **86.7** | 80.0 | 66.7 | 66.7 | 0.46 | 0.50 | 0.56 | 0.42 |
> |  | Skywork-OR1-7B / 32B | **86.7** | 80.0 | 60.0 | 73.3 | **0.33** | **0.27** | **0.19** | **0.24** |
> | **AIME25** | Qwen2.5-7B / 32B | **40.0** | 33.3 | 26.7 | **40.0** | 0.62 | 0.68 | 0.70 | 0.74 |
> |  | s1.1-7B / 32B | **53.3** | **53.3** | 33.3 | 40.0 | 0.49 | 0.53 | 0.64 | 0.52 |
> |  | DeepSeek-R1-1.5B / 32B | **60.0** | 53.3 | 46.7 | 35.7 | 0.47 | 0.50 | 0.55 | 0.43 |
> |  | Skywork-OR1-7B / 32B | **60.0** | 53.3 | 40.0 | 53.3 | **0.41** | **0.36** | **0.29** | **0.22** |
> | **AMC23** | Qwen2.5-7B / 32B | 72.0 | 70.0 | **74.0** | 72.0 | 0.63 | 0.69 | 0.71 | 0.75 |
> |  | s1.1-7B / 32B | **82.0** | 78.0 | 76.0 | 78.0 | 0.48 | 0.52 | 0.62 | 0.48 |
> |  | DeepSeek-R1-1.5B / 32B | **92.0** | 88.0 | 82.0 | 80.0 | 0.46 | 0.50 | 0.57 | 0.44 |
> |  | Skywork-OR1-7B / 32B | **96.0** | **94.0** | 82.0 | 86.0 | **0.39** | **0.34** | **0.37** | **0.28** |
>
>
>
> **A5:** Regarding the empirical evidence of its superiority, as shown in Table 2 of our paper, we have compared our method against theirs and demonstrated not only higher inference speedups but also lower token consumption. Moreover, in most cases, our approach surpasses their performance.

---

> ### Author Response · Authors · 2025-11-13
> **Response to Reviewer ct8q Part VI**
>
> **Q6:** System/engineering considerations The paper does not discuss the impact of modern inference engine features, such as supporting asynchronous verification requests. Can ATTS be integrated with inference backends that support asynchronous calls, and is there any experimental evidence for further gains from such integration?
>
> **A6:** Thank you for your valuable feedback.
>
> - Regarding supporting asynchronous verification requests, sglang inherently has the capability to handle requests asynchronously and can support streaming input. Its maximum number of concurrent verification requests is determined solely by the number of computing cores on the deployment server. However, a large number of concurrent requests significantly increases memory pressure, raising the risk of memory overflow errors in sglang. Therefore, after completing the prefill stage (i.e., verification), we prioritize decoding tasks (target model sampling) for high-priority requests. Before transitioning from the prefill stage (target model verification) to the decoding stage (target model decoding), requests are sorted, which introduces a synchronization operation.
>
> - Regarding “integration with inference backends,” our current scheduling strategy is implemented on the frontend (scheduler).  It is important to note that sglang’s frontend handles request scheduling features such as prefix sharing. I believe that performing task scheduling for target model sampling in rejection sampling on the frontend is sufficient, as it can reduce intrusive modifications to the inference engine and avoid degrading the performance of the existing foundational setup.
>
> - The backend of sglang, on the other hand, mainly deals with the model’s computational operators. If the internal computation pipelines of different requests involve synchronization or ordering dependencies — for example, the well-known branch prediction problem[1,2,3] — The core idea of ATTS can be applied to effectively address and manage these branch prediction problems. Since our main improvements focus on the infrastructure scheduling layer, we did not further extend the experiments to the infrastructure backend.
>
>
>
>
> **Reference:**
> [1] Dynamic Branch Prediction with Perceptrons.  2001, HPCA
> [2] A Study of Branch Prediction Strategies. 1981, ISCA
> [3] BranchNet: A Convolutional Neural Network to Predict Hard-To-Predict Branches. 2020, MICRO

---

> > ### Author Response · Authors · 2025-11-18
> > **Humble Request for Consideration**
> >
> > If you find our results helpful for your evaluation, we would sincerely appreciate it if you could kindly consider giving us a higher score.

---

> ### Author Response · Authors · 2025-11-24
> **Experiments in Multi-Task Settings**
>
> **Table3**: Marginal  Coverage (MargCov ) by combining data and calibration sets from both tasks and Conditional Coverage (CondCov Rate 2) by Task Type (Math & AIME25); 15 turns, budget = 500, error = 0.2.
>
> | Draft Model                          | Target Model             | Dataset | CondCov  — Conditional | MargCov - Mixed |
> |--------------------------------------|---------------------------|----------|-------------------------------|-----------------------------|
> | Qwen/Qwen2.5-7B-Instruct             | simplescaling/s1.1-32B    | aime25   | 33.33                         | 26.67                      |
> | Qwen/Qwen2.5-7B-Instruct             | simplescaling/s1.1-32B    | math500  | 90                            | 89                         |
> | deepseek-ai/DeepSeek-R1-Distill-Qwen-1.5B | simplescaling/s1.1-32B    | aime25   | 60                            | 53.33                      |
> | deepseek-ai/DeepSeek-R1-Distill-Qwen-1.5B | simplescaling/s1.1-32B    | math500  | 92                            | 92                         |
> - According to your suggestion, we have added experiments under a multi-task setting, which better reflects the requirements of real-world deployment. In this setup, we compare two configurations: one that makes no assumptions about the task type of incoming requests (marginal coverage), and another that assumes knowledge of the task type for each incoming request (conditional coverage). **These two settings simulate different real-world scenarios where requests from multiple task types arrive in a streaming manner.**
> - We observe that when the task type of each request is available, computing the p-value conditionally based on the corresponding calibration set yields clearly better performance on both the MATH and AIME25 datasets.
> - For the relatively simpler MATH dataset, conditioning on the known task type yields only marginal changes in performance. In contrast, for the more challenging AIME25 dataset, allocating the target model’s budget precisely based on the task type leads to a substantial improvement in final performance.
> - We sincerely thank you again for your time and effort in reviewing our work. If our additional experiments have addressed your concerns, we would greatly appreciate it if you could consider adjusting your score, or we would be happy to further discuss any remaining questions.

---

> ### Author Response · Authors · 2025-11-27
> **Kindly Requesting Response Before Deadline**
>
> Dear Reviewer,
>
> We sincerely hope this message finds you well. As the ICLR rebuttal deadline is approaching, we wanted to kindly check whether you might have a chance to respond to our clarification questions. We fully understand that you must be extremely busy, but your feedback would be invaluable for us to improve and better address your concerns before the final submission.
>
> We deeply appreciate your time and effort in reviewing our paper, and thank you again for your thoughtful comments and suggestions.
>
> Warm regards,
> Authors

---

### Official Review · Reviewer_KbTN · 2025-10-31

**Soundness:** 3
**Presentation:** 3
**Contribution:** 3
**Rating:** 8
**Confidence:** 2

**Summary:**

This paper proposes ATTS, an asynchronous test-time scaling framework for large language models (LLMs) that addresses high inference latency and memory bottlenecks in test-time scaling. By redefining asynchronous arithmetic intensity to identify synchronization as the key bottleneck, integrating online calibration with conformal prediction, and designing a three-stage rejection sampling pipeline, ATTS enables both sequential and parallel scaling. Experiments on MATH, AMC23, AIME24/25 datasets show up to 56.7x speedup, 4.14x throughput improvement, and no accuracy loss—even allowing 1.5B/70B draft/target models to match o3-mini (high) performance on AIME.

**Strengths:**

1. The introduction of "asynchronous arithmetic intensity" effectively quantifies synchronization overhead, a critical but understudied issue in test-time scaling.
2. Leveraging conformal prediction for ranking and rejection sampling ensures controlled rejection rates and coverage (marginal/conditional), adding theoretical rigor absent in many efficiency-focused works.
3. Significant speedup and throughput gains across model families (e.g., Qwen, DeepSeek, Llama) and challenging math benchmarks demonstrate real-world utility.

**Weaknesses:**

1. ATTS frames sequential scaling (increasing turns) as a strength, but experiments only test up to 10 turns. The paper fails to analyze when sequential scaling stops adding value—e.g., whether 20+ turns lead to accuracy plateaus while inflating token costs.
2. The evaluation on more challnaging benchmarks would make the results more robust
3. While conformal prediction enables statistical guarantees for rejection sampling, the paper overlooks a critical tension: its p-value calculation (Eq.13) requires comparing test samples to the full calibration set. For large batch sizes (128+), this comparison could introduce hidden per-sample latency—undermining ATTS’s core goal of reducing overhead.

**Questions:**

See weakness

---

> ### Author Response · Authors · 2025-11-17
> **Response to Reviewer ct8q Part I**
>
> **Q1:** The evaluation on more challenging benchmarks would make the results more robust.
>
> **A1:** Thank you for your suggestion. We have added results on the more challenging *OlympiadBench* benchmark \[1].
>
> | Draft Model                               | Target Model           | Draft Model Baseline | Target Model Baseline | Marginal Coverage | Conditional Coverage |
> | ----------------------------------------- | ---------------------- | -------------------- | --------------------- | ----------------- | -------------------- |
> | deepseek-ai/DeepSeek-R1-Distill-Llama-8B  | simplescaling/s1.1-32B | 28                   | 48                    | 44                | 40                   |
> | deepseek-ai/DeepSeek-R1-Distill-Qwen-1.5B | simplescaling/s1.1-32B | 26                   | 48                    | 38                | 38                   |
> | meta-llama/Llama-3.1-8B-Instruct          | simplescaling/s1.1-32B | 26                   | 48                    | 44                | **50**               |
> | Qwen/Qwen2.5-7B-Instruct                  | simplescaling/s1.1-32B | 32                   | 48                    | 48                | 48                   |
> | deepseek-ai/DeepSeek-R1-Distill-Llama-8B  | Qwen/QwQ-32B           | 28                   | 46                    | 38                | 40                   |
> | deepseek-ai/DeepSeek-R1-Distill-Qwen-1.5B | Qwen/QwQ-32B           | 26                   | 46                    | 38                | 40                   |
> | meta-llama/Llama-3.1-8B-Instruct          | Qwen/QwQ-32B           | 26                   | 46                    | 48                | **48**               |
> | Qwen/Qwen2.5-7B-Instruct                  | Qwen/QwQ-32B           | 32                   | 46                    | 50                | 46                   |
>
> **Table 1.** The setting involves a sample size of 16 for each turn, with each sample undergoing 15 turn. The value of α is set to 0.4. For each round, the token budget is allocated as 500 for both the target model and the draft model. The sampling temperature is set to 0.8.
>
> - We chose the more challenging *OlympiadBench* benchmark, and under two of our settings (results shown in **bold**), we were even able to surpass the performance of the original target model’s sampling.
> - Under conditional coverage, we achieved a more efficient allocation of computational resources compared to marginal coverage, resulting in improved test performance.
>
>
> **Reference:**
> \[1] *OlympiadBench: A Challenging Benchmark for Promoting AGI with Olympiad-Level Bilingual Multimodal Scientific Problems.* ACL 2024.

---

> ### Author Response · Authors · 2025-11-17
> **Response to Reviewer ct8q Part II**
>
> **Q2:** While conformal prediction enables statistical guarantees for rejection sampling, the paper overlooks a critical tension: its p-value calculation (Eq.13) requires comparing test samples to the full calibration set. For large batch sizes (128+), this comparison could introduce hidden per-sample latency—undermining ATTS’s core goal of reducing overhead.
>
> **A2:** In fact, in the code implementation, the entire calibration set is stored as an array of floating-point numbers. Comparing a PPL of test sample with the entire calibration set only requires an O(n) algorithm, whose computational complexity is negligible compared to the overhead of other matrix multiplications and synchronization operations.

---

> ### Author Response · Authors · 2025-11-18
> **Response to Reviewer ct8q Part III**
>
> **A3:** ATTS frames sequential scaling (increasing turns) as a strength, but experiments only test up to 10 turns. The paper fails to analyze when sequential scaling stops adding value—e.g., whether 20+ turns lead to accuracy plateaus while inflating token costs.
>
> **Q3:**  Thank you for your suggestion. We provide the following experimental results in AIME25. In this setting, we use a sample size of 16, a significance level of 0.4, and allocate a budget of 400 for both the target model and the draft model in each turn, with the temperature set to 0.8.
>
>
> | Draft Model | Target Model | Turn 2 | Turn 4 | Turn 8 | Turn 12 | Turn 16 | Turn 20 | Turn 24 | Turn 28 |
> |-------------|--------------|--------|--------|--------|---------|---------|---------|---------|---------|
> | deepseek-ai/DeepSeek-R1-Distill-Qwen-1.5B | simplescaling/s1.1-32B | 0 | 13.33 | 33.33 | 33.33 | 46.67 | 60 | 60 | 60 |
> | Qwen/Qwen2.5-7B-Instruct | simplescaling/s1.1-32B | 13.33 | 33.33 | 40 | 40 | 40 | 40 | 40 | 40 |
> **Table1:** Marginal Coverage
>
>
> | Draft Model | Target Model | Turn 2 | Turn 4 | Turn 8 | Turn 12 | Turn 16 | Turn 20 | Turn 24 | Turn 28 |
> |-------------|--------------|--------|--------|--------|---------|---------|---------|---------|---------|
> | deepseek-ai/DeepSeek-R1-Distill-Qwen-1.5B | simplescaling/s1.1-32B | 6.67 | 26.67 | 40 | 46.67 | 46.67 | 53.33 | 60 | 60 |
> | Qwen/Qwen2.5-7B-Instruct | simplescaling/s1.1-32B | 20 | 26.67 | 26.67 | 40 | 40 | 40 | 40 | 40 |
> **Table2:** Conditional Coverage
>
>
> - Our experiments demonstrate the results of scaling the number of turns to more than 20, showing that the performance reaches its peak after 24 turns, with no additional gains beyond that point.
> - There is a slight difference from the original setup: in each turn we use a smaller token budget (400), but more turns, we did not surpass the peak performance reported in the original paper. Therefore, when the number of sampling of each turn is fixed, scaling the number of turns has an upper limit.
>
> **Reference:**
> [1]  s1: Simple test-time scaling. 2025, EMNLP
>
> **If you find our results helpful for your review, we would be deeply grateful if you could consider giving us a higher score**.

---

> ### Author Response · Authors · 2025-11-28
> **Follow-up on Review Feedback for Submission 10643**
>
> Dear Reviewer KbTN,
>
> I hope this message finds you well. The rebuttal deadline for our submission “Asynchronous Test-Time Scaling (ATTS)” is approaching soon. We truly appreciate your thoughtful review and detailed comments, which have been invaluable to our revisions.
>
> If you have any further feedback or clarifications you’d like us to address before the rebuttal deadline, we would be very grateful to receive them. Your additional insights would help us ensure that our responses are as clear and complete as possible.
>
> Thank you again for your time and constructive evaluation.
>
> Warm regards,
> Authors

---

### Official Review · Reviewer_LsxV · 2025-11-01

**Soundness:** 3
**Presentation:** 3
**Contribution:** 3
**Rating:** 8
**Confidence:** 3

**Summary:**

The authors propose ATTS (Asynchronous Test-Time Scaling), designed to alleviate the memory peak and synchronization overhead issues in speculative decoding under parallel and sequential test-time scaling settings.

The authors characterize system bottlenecks using asynchronous arithmetic intensity, explicitly incorporating synchronization time into the performance metric. The authors find that they synchronous time is the main bottleneck. Therefore, ATTS adopts conformal prediction: it computes a p-value for each candidate and compares it with a threshold α to decide acceptance or rejection, instead of using global softmax and ranking which needs synchronous. This provides statistical control over the target model intervention rate without requiring distributional assumptions.

Building on this principle, ATTS implements a three-stage rejection sampling pipeline that scales along both the parallel and sequential axes. It further employs online calibration and ordered decision-making to avoid global synchronization.

Experiments on MATH, AMC23, AIME24/25, and various draft/target model combinations show that ATTS achieves up to 56.7× end-to-end acceleration and 4.14× throughput improvement without sacrificing accuracy, while significantly reducing peak memory usage and latency. Notably, on the AIME benchmark, a 1.5B draft / 70B target setup reaches performance close to that of strong closed-source reasoning models.

The main limitations lie in the sensitivity to online calibration quality and the choice of α, the lack of guarantees for global optimality in each round, and the evaluation focus on mathematical reasoning and specific inference engine configurations.

**Strengths:**

1. This paper addresses a highly important and practical problem in inference infrastructure, which has significant implications for optimizing the efficiency of model deployment.
2. The authors introduced a new performance metric to analyze specific bottlenecks, which guided the subsequent algorithm design.
3. The paper is well-structured and easy to understand.
4. The proposed method achieves a significant improvement in efficiency while maintaining stable performance.

**Weaknesses:**

1. The proposed method heavily relies on the stability of online calibration. The rejection rate control and accuracy are quite sensitive to the calibration distribution, the number of parallel samples, and the value of α.
2. The proposed method does not aim for globally optimal candidates and cannot guarantee obtaining the top-1 candidate in each round, which may be suboptimal for applications that strictly require optimal ranking.

**Questions:**

How can online calibration maintain long-term stability of the rejection rate under non-stationary workloads?

---

> ### Author Response · Authors · 2025-11-13
> **Response to Reviewer LsxV**
>
> **Q1**: The proposed method heavily relies on the stability of online calibration. The rejection rate control and accuracy are quite sensitive to the calibration distribution, the number of parallel samples, and the value of α.
>
>
> **A1**:  Thank you for your suggestion.
> - I would like to clarify one concept: in this paper, the value of α is kept consistent between the family-wise error rate and the rejection rate in conformal prediction. Intuitively, a higher rejection rate means that the target model is invoked more frequently during test-time scaling, which increases the usage of computational and memory resources but generally leads to better performance.
> - If more precise control over the rejection rate is desired, we can follow the common practice in conformal prediction and specify a smaller α value to mitigate the risk of load imbalance caused by excessively high rejection rates.
> - According to the results shown in Figure 6 of the paper, when the sample size is sufficiently large, the prediction error of our budget can be controlled within 2%. Based on the classical Central Limit Theorem, the larger the scaling sample size, the more accurate our prediction becomes. When the sample size is small, due to the ample computational and memory resources, the target model can appropriately handle requests that exceed the rejection rate, and in such cases, the resulting error is acceptable.
>
>
>
> **Q2**: The proposed method does not aim for globally optimal candidates and cannot guarantee obtaining the top-1 candidate in each round, which may be suboptimal for applications that strictly require optimal ranking.
>
>
> **A2**: How to obtain the optimal candidate (ground truth) in each round is indeed an interesting question. Similar to the goal in the classical literature [1,2], which seeks to achieve conditional coverage within an error rate α such that the ground truth falls within the prediction set, we need to design stricter conformal p-values to obtain this theoretical guarantee. This poses some new theoretical challenges, which we plan to explore in future work.
>
>
> **Q3**: How can online calibration maintain long-term stability of the rejection rate under non-stationary workloads?
>
> **A3:** Thank you for your insightful feedback.
>
>
> - My understanding of "non-stationary workloads" refers to situations where the input data distribution changes over time, such as peaks in business traffic, shifts in user behavior, or seasonal variations. I believe there are several possible directions for improvement.
>
>
> - When the data distribution changes over time, we can consider updating our calibration set accordingly — that is, re-sampling calibration examples from the shifted distribution to adapt to the distributional drift.
>
>
>
>
> **Reference**:
> [1] Classification with valid and adaptive coverage. 2020, NeurIPS
> [2] Class-Conditional Conformal Prediction with Many Classes.  2023, NeurIPS

---

> > ### Author Response · Authors · 2025-11-18
> > **Humble Request for Consideration**
> >
> > If you find our results helpful for your evaluation, we would sincerely appreciate it if you could kindly consider giving us a higher score.

---

> ### Author Response · Authors · 2025-11-28
> **Follow-up on Review Feedback for Submission 10643**
>
> Dear Reviewer LsxV,
>
> I hope this message finds you well. The rebuttal deadline for our submission “Asynchronous Test-Time Scaling (ATTS)” is approaching soon. We truly appreciate your thoughtful review and detailed comments, which have been invaluable to our revisions.
>
> If you have any further feedback or clarifications you’d like us to address before the rebuttal deadline, we would be very grateful to receive them. Your additional insights would help us ensure that our responses are as clear and complete as possible.
>
> Thank you again for your time and constructive evaluation.
>
> Warm regards,
> Authors

---

### Author Response · Authors · 2025-11-21
**Summary of Revisions to the ATTS**

Dear Reviewers and Area Chair,

We have updated our revised PDF with several new additions and clarifications:

New Experimental Results: We have included additional experiments to further validate our findings.

Discussion on the Assumption of Probability Normalization (Section 3.2): We now cite prior research related to the assumption that the output probabilities of weakened models sum to one, and we expand our discussion on related work in conformal p-value studies.

Results on the OlympiadBench Dataset (Section 4.5): We have added experiments conducted on the OlympiadBench benchmark.

Experiments with More Iteration Turns (Section 4.6): We report results under larger numbers of reasoning iterations.

Discussion of Asynchronous Features in Modern Inference Engines (Section 4.7): We added a dedicated section discussing the asynchronous nature of modern reasoning engines and how they encounter synchronization bottlenecks.

**Thank you again for the time and effort you put into reviewing our manuscript. If our clarifications have been helpful, we kindly ask you to consider raising our score. If you have any additional questions, we would be happy to discuss them further.** Thank you very much for your time and consideration.

Best regards,
Authors

---

### Meta-Review · Area_Chair_CgdB · 2026-01-07

**Summary:**

This paper proposes ATTS (Asynchronous Test-Time Scaling), a system-level framework for accellerating test-time scaling by removing synchronization bottlenecks via asynchronous inference, conformal prediction, and online calibration. The core contribution is to replace synchronized ranking and softmax-based selection with a statistically grounded rejection mechanism that supports both parallel and sequential scaling.

Two reviewers find the contribution important, novel, and practically impactful, with strong empirical evidence of large speedups and throughput gains without accuracy loss. One reviewer assigns a low score, but their concerns seem to largely stem from factual misunderstandings or issues already addressed in the submission and rebuttal. Overall, after considering the rebuttal and the balance of reviews, the paper appears borderline but leaning accept, especially given its strong systems relevance and resolved concerns.

**Reviewer Concerns:**

Concerns fully or mostly addressed by the rebuttal:
* Missing baselines: Comparisons to SpecReason and Speculative Thinking were already present in the original submission and were explicitly reiterated and expanded during rebuttal with detailed tables.
* Distribution mismatch concerns: Additional experiments demonstrate that ATTS maintains strong speedups (e.g., >20×) even when draft and target models are trained on different distributions, contradicting the claim that benefits diminish under mismatch.
* Theoretical validity of conformal prediction without normalization: The authors clarified that ATTS uses conformal p-values rather than normalized probabilities, provided theoretical justification (including references and proofs), and showed that exchangeability assumptions still hold.
* System integration and asyncrony: The authors added discussion and clarification on compatibility with modern inference engines and asynchronous scheduling, addressing system-level deployment questions.
* Multi-task and large-scale deployment: Additional multi-task experiments with marginal vs. conditional calibration were included, showing improved performance under task-aware calibration.

Concerns that remain mostly/fully outstanding:
* Sensitivity to calibration and α: While empirically explored and shown to be stable within reasonable regimes, the method’s guarantees remain marginal/conditional rather than globally optimal, which the authors acknowledge as a limitation and future work.
* Eval Scope: Experiments focus on mathematical reasoning benchmarks, broader task diversity (eg, non-math reasoning) would further strengthen generality, though this is not atypical for work in test-time scaling.

**Reviewer Scores:**

* Reviewer KbTN: Likely remains at 8. Their concerns about scaling limits and calibration cost were directly addressed with new experiments and analysis.
* Reviewer LsxV: Likely remains at 8. The rebuttal clearly addressed calibration stability, α sensitivity, and system integration questions.
* Reviewer ct8q (2: reject): Likely increases to 4 (perhaps even 6) under good-faith discussion. Core objections (missing baselines, unfair comparisons, invalid assumptions) were factually incorrect or fully resolved by existing tables and new rebuttal material; remaining concerns align more with acknowledged limitations than fatal flaws.

---

### Decision · Program_Chairs · 2026-01-26

Accept (Poster)